# Revisiting Positive Samples in Graph Contrastive Learning:
# From the Perspective of Message Passing

Lianze Shan [† 1]   Ningchong Wang [† 2]   Jitao Zhao [† 1]   Di Jin [1]   Dongxiao He [* 1]

## Abstract

Graph Contrastive Learning (GCL), which trains graph encoders by maximizing similarity between positive samples and minimizing it between negative ones, has emerged as a mainstream graph pre-training paradigm. It is widely recognized that positive samples are essential in GCLs. Ideally, maximizing the similarity of positive samples enables graph encoders to capture intrinsic semantics and patterns of graph data. However, we discover an interesting phenomenon: *GCLs can achieve competitive performance even without positive samples.* This motivates us to revisit the fundamental mechanism of positive samples in GCLs. From the perspective of Dirichlet energy, we theoretically find that message passing, a key mechanism in graph encoders, trivializes the maximization of positive samples, preventing GCLs from effectively learning from positive samples. To address this, we propose SPGCL to mitigate the trivialization caused by message passing and restore the learning efficacy of positive samples. Specifically, we find that high Dirichlet energy features help positive samples provide effective learning signals while low Dirichlet energy features contribute little to positive learning signal but is useful for positive sampling. Based on this, SPGCL introduces an energy-aware propagation mechanism that selectively propagates features based on their Dirichlet energy, preventing low-energy features from dominating similarity. Furthermore, we construct a positive sampling matrix based on Dirichlet energy to exclude misleading positive samples and focus on node pairs with higher discrimination. Extensive experiments demonstrate the effectiveness of SPGCL.

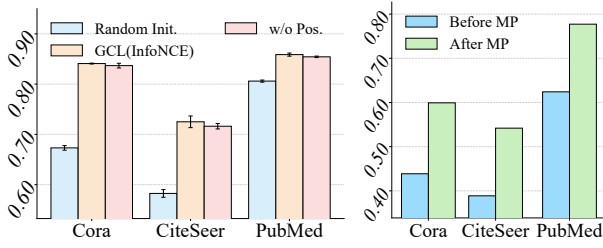

*(a)* Impact of positive alignment.    *(b)* Pos. similarity.

*Figure 1.* Motivation experiments (Acc.). In (a), Random Init. means a GCN encoder with randomly initialized parameters without any training; GCL (InfoNCE) means a standard graph contrastive learning model trained with the InfoNCE loss; w/o Pos. means a GCL variant where the positive alignment term is removed from the InfoNCE loss. In (b), Before MP means the average cosine similarity of positive samples before message passing; After MP means the average cosine similarity after message passing.

## 1. Introduction

Graph Contrastive Learning (GCL), which trains Graph Neural Network (GNN) encoders by aligning positive samples while distinguishing them from negative ones, has emerged as a mainstream self-supervised pre-training paradigm for learning graph representations without labels (Wu et al., 2021a). It has been extensively deployed across diverse real-world scenarios and achieved great success, including e-commerce recommendation systems (He et al., 2020), social network analysis (Yanardag & Vishwanathan, 2015), and molecular property prediction (Li et al., 2022).

In GCLs, positive samples are widely regarded as a key component (Thakoor et al., 2021; Liu et al., 2022a; Lee et al., 2022; Li et al., 2023). GCLs typically construct positive samples by augmentation-based sampling (Zhu et al., 2020; Velickovic et al., 2019; Zhu et al., 2021) or neighborhood-based sampling (Zhang et al., 2022a; He et al., 2023). Ideally, aligning positive pairs enables graph encoders to learn invariant representations that encode intrinsic semantic and structural patterns of graph data (Zhu et al., 2021). A broad consensus holds that the efficacy of GCLs critically depends on the quality of learning from this alignment process. Therefore, many recent works focus on designing various augmentation and sampling strategies to enhance the ef-

[†]Equal contribution. [*]Corresponding author. [1]School of Computer Science and Technology, Tianjin University, Tianjin, China [2]School of Future Technology, Tianjin University, Tianjin, China. Correspondence to: Dongxiao He <hedongxiao@tju.edu.cn>.

*Proceedings of the 43ʳᵈ International Conference on Machine Learning*, Seoul, South Korea. PMLR 306, 2026. Copyright 2026 by the author(s).

fectiveness of learning from positive samples. For example, GCA (Zhu et al., 2021), SpCo (Liu et al., 2022a), and CSGCL (Chen et al., 2023) design effective augmentation strategies to improve the construction of positive samples. LocalGCL (Zhang et al., 2022a), AFGRL (Lee et al., 2022), and NeCo (He et al., 2023) improve positive samples by adopting carefully designed neighbor sampling.

However, we find a counterintuitive phenomenon: *the positive samples, which are widely considered essential in GCLs, have surprisingly limited impact on performance.* Here, we begin with a motivation experiment and the results are presented in Figure 1 (a). We show the results of three models: a randomly initialized Graph Convolutional Network (GCN) (Kipf & Welling, 2017), a GCL model trained with the InfoNCE loss (Zhu et al., 2020), and a GCL variant without positive samples. It can be observed that removing positive samples achieves competitive performance, closely matching the standard GCL and outperforming the randomly initialized GCN. This suggests that the benefit of positive alignment is limited, prompting us to ask: **Why** do positive samples fail to provide significant performance gains? **What** constrains learning from positive samples? **How** to learn more effectively from positive samples? Essentially, the effectiveness of positive samples stems from learning invariant semantics under substantial perturbations; i.e., positive samples should exhibit significant diversity while preserving core semantics. However, as shown in Figure 1(b), we observe a marked increase in the average representation similarity of positive samples after message passing. This motivates us to theoretically investigate the impact of message passing on positive samples learning. Consistent with our empirical observation, we theoretically find that message passing increases the expected similarity between positive samples. This reduces the diversity between positive samples before contrastive optimization, which we refer to as a pre-alignment effect, weakening the learning signal carried by positive samples.

We present a detailed theoretical analysis in Section 3. Specifically, we first analyze how GNN message passing affects positive samples under two widely used positive construction strategies, including augmentation-based positive samples and neighborhood–based positive samples. We theoretically prove that GNN message passing inherently increases the expected similarity between positive samples, implying that positive samples can become overly similar before contrastive optimization. Consequently, similarity increase alone can not distinguish effective positive alignment from trivial similarity induced by message passing, and thus is insufficient to reflect how much useful information GCL can learn from positive samples. We further introduce a metric to measure the effectiveness of positive sample learning and define it as the mutual information between the ground truth sample distribution and the embedding similarity distribution. We then prove that this effectiveness is strictly upper bounded by the Dirichlet energy of each feature, i.e., low-energy features dominate the similarity score but offer negligible signals for positive samples learning. This not only explains why GCLs maintain competitive performance without positive samples, but also provides a principled guideline for optimizing positive samples learning in GCLs.

Based on the above findings and theoretical analysis, we propose **S**eparate **P**ropagation **G**raph **C**ontrastive **L**earning (SPGCL) [1] to reduce the pre-alignment effect of message passing and better exploit positive samples. Specifically, SPGCL introduces an Energy Aware Propagation (EAP) mechanism. For each feature dimension, we estimate its Dirichlet energy and separate its propagation accordingly. Low-energy features are excluded from message passing to eliminate their contribution to redundant pre-alignment effect, while high-energy features are propagated to preserve informative diversity. As a result, contrastive objective is forced to capture invariant representations from feature components that provide effective positive learning signals. Furthermore, we design an Energy-guided Positive Sampling (EPS) mechanism to construct more reliable positive samples. Existing GCL methods typically rely on heuristic data augmentations (Zhu et al., 2020; Velickovic et al., 2019; Thakoor et al., 2021) or local neighborhood–based sampling (Zhang et al., 2022a; Lee et al., 2022) to define positive samples. These strategies inevitably introduce misleading positive samples, as stochastic perturbations may destroy core semantics while local neighbors lack semantic consistency in heterophilic graphs. Therefore, we leverage low-energy features to construct a probabilistic positive sampling matrix. Since low-energy features capture stable signals that are less sensitive to perturbations and neighborhood noise, using them for positive sampling helps filter out misleading positives while retaining semantically reliable relations. Our contributions can be summarized as follows:

- We find a phenomenon in GCLs, termed pre-alignment effect, where message passing makes positive samples similar before optimization, weakening the effective learning from positive samples.

- We propose SPGCL, which restores informative positive learning by separating feature propagation based on Dirichlet energy and using low-energy features to guide positive sampling.

- We conducted extensive experiments on 12 graph benchmarks, including node classification and clustering tasks. The results demonstrate that SPGCL outperforms state-of-the-art GCLs.

---

[1]Code is publicly available at https://github.com/hedongxiao-tju/SPGCL.

## 2. Preliminary

In this section, we introduce the basic concepts and notations used in this paper, including graph-structured data, graph neural networks, and graph contrastive learning. Unless otherwise specified, we use calligraphic symbols to denote sets, bold uppercase letters for matrices, bold lowercase letters for vectors, and lowercase letters for scalars.

**Graph-structured data.** A graph is denoted as $\mathcal{G} = (\mathcal{V}, \mathcal{T})$, where $\mathcal{V} = \{v_1, \ldots, v_N\}$ is the set of $N$ nodes and $\mathcal{T} \subseteq \mathcal{V} \times \mathcal{V}$ is the set of edges. Each node $v_i \in \mathcal{V}$ is associated with a feature vector $\mathbf{x}_i \in \mathbb{R}^F$, and all node features are stacked into a feature matrix $\mathbf{X} \in \mathbb{R}^{N \times F}$. The graph structure is represented by an adjacency matrix $\mathbf{A} \in \{0, 1\}^{N \times N}$, where $\mathbf{A}_{ij} = 1$ if $(v_i, v_j) \in \mathcal{T}$ and 0 otherwise.

**Graph Neural Network.** Graph Neural Networks (GNNs) aim to learn node representations by iteratively aggregating information from local neighbors (Wu et al., 2021b; Zhou et al., 2020). Most GNNs follow the message passing framework, where the representation of each node is updated by combining its own features with its neighbors. Formally, let $\mathbf{H}^{(0)} = \mathbf{X}$ denote the input node features, at the $l$-th layer, a GNN encoder updates node representations as:

$$\mathbf{H}^{(l+1)} = \sigma\left(\mathbf{P}\mathbf{H}^{(l)}\mathbf{W}^{(l)}\right), \tag{1}$$

where $\mathbf{W}^{(l)}$ is a learnable weight matrix, $\sigma(\cdot)$ is a nonlinear activation function, and $\mathbf{P}$ is a propagation operator derived from the graph structure. A representative example of GNNs is Graph Convolutional Network (GCN) (Kipf & Welling, 2017), which employs the normalized adjacency matrix with self-loops for propagation:

$$\mathbf{P} = \tilde{\mathbf{D}}^{-\frac{1}{2}}\tilde{\mathbf{A}}\tilde{\mathbf{D}}^{-\frac{1}{2}}, \tilde{\mathbf{A}} = \mathbf{A} + \mathbf{I}, \tag{2}$$

where $\tilde{\mathbf{D}}$ is the corresponding degree matrix. This propagation can be viewed as a form of Laplacian smoothing that encourages neighboring nodes to have similar representations. This smoothing effect can be measured by the Dirichlet energy. Given a graph signal $\mathbf{f} \in \mathbb{R}^N$, its Dirichlet energy is defined as:

$$\mathcal{E}(\mathbf{f}) = \mathbf{f}^\top \mathbf{L}\mathbf{f} = \frac{1}{2}\sum_{(v_i, v_j) \in \mathcal{T}} \mathbf{A}_{ij}(f_i - f_j)^2, \tag{3}$$

where $\mathbf{L} = \mathbf{D} - \mathbf{A}$ is the graph Laplacian. Low-energy features correspond to smoother signals across the graph, while high-energy features preserve greater variation among neighboring nodes.

**Graph Contrastive Learning.** Graph Contrastive Learning (GCL) aims to train GNN encoders based on mutual information maximization (Velickovic et al., 2019). However, directly maximizing mutual information is intractable in practice. Instead, most GCL methods adopt contrastive objectives that provide a lower bound on mutual information. A commonly used formulation is the InfoNCE loss (Oord et al., 2018), defined as:

$$\mathcal{L}_{\text{InfoNCE}}(\mathbf{h}_i) = -\log\bigg($$
$$\frac{\sum_{p \in \mathcal{P}_i} \exp(\text{sim}(\mathbf{h}_i, \mathbf{h}_p)/\tau)}{\sum_{p \in \mathcal{P}_i} \exp(\text{sim}(\mathbf{h}_i, \mathbf{h}_p)/\tau) + \sum_{n \in \mathcal{N}_i} \exp(\text{sim}(\mathbf{h}_i, \mathbf{h}_n)/\tau)}\bigg), \tag{4}$$

where $\mathcal{P}_i$ and $\mathcal{N}_i$ denote the sets of positive and negative representations for anchor $\mathbf{h}_i$, respectively, $\text{sim}(\cdot, \cdot)$ denotes a similarity function, $\tau$ is a temperature parameter, and $\exp(\cdot)$ denotes the exponential function.

## 3. Theoretical Analysis

In this section, we theoretically investigate how GNN message passing affects positive samples learning in graph contrastive learning. We first show that, under common positive samples construction rules, message passing inherently increases the expected similarity between positive samples, leading to a pre-alignment effect. This observation indicates that increased similarity alone is insufficient to characterize how much information GCL actually learns from positive alignment. To address this, we introduce an information-theoretic metric to quantify positive learning effectiveness, defined as the mutual information between the ground truth sample distribution and the embedding similarity encoded by GNN. Based on this, we further derive a feature-wise upper bound, showing that the contribution of each feature dimension to positive learning effectiveness is limited by its Dirichlet energy. These results reveal that low-energy features dominate similarity increase while providing limited learning signal, which motivates separating their propagation to better exploit informative positive alignment.

We now analyze the effect of message passing on positive samples under common positive construction rules and give the following lemma:

**Lemma 3.1** (Pre-alignment effect of Message Passing). *Let* $\mathbf{P}$ *be a symmetric propagation operator and denote* $\mathbf{H}^{(l+1)} \triangleq \mathbf{P}^{(l)}\mathbf{H}^{(l)}$. *For a node pair* $(v_i, v_j)$, *let* $R_{ij}^{(l)} \triangleq \phi\left(\mathbf{H}_{i,:}^{(l)}, \mathbf{H}_{j,:}^{(l)}\right)$, *where* $\phi$ *is an inner product or cosine similarity with normalized features. Then, under common positive construction rules, for* $\forall l \geq 0$ *we have:*

$$\mathbb{E}\left[R_{ij}^{(l+1)} \mid (v_i, v_j) \in \mathcal{P}\right] \geq \mathbb{E}\left[R_{ij}^{(l)} \mid (v_i, v_j) \in \mathcal{P}\right]. \tag{5}$$

This effect holds under two commonly used positive construction rules in GCLs: augmentation-based positive samples and neighborhood-based positive samples. We provide

detailed proofs in Appendix E.1. Lemma 3.1 shows that message passing increases the expected similarity of positive samples. Consequently, the increase of similarity itself can not distinguish informative positive alignment from trivial similarity induced by message passing, and is therefore insufficient to characterize how much information GCL can learn from positive samples. To quantify this, we introduce the following definition:

**Definition 3.2** (Positive Samples Learning Effectiveness). Let $S_{ij} \in \{0, 1\}$ be a binary random variable indicating whether a node pair $(v_i, v_j)$ is generated by the positive samples construction rule, where $S_{ij} = 1$ denotes a positive sample and $S_{ij} = 0$ denotes a random sample. Let $R_{ij}$ denote the embedding similarity score produced by a GNN encoder for the pair $(v_i, v_j)$. The *positive samples learning effectiveness* is defined as:

$$\mathcal{I} \triangleq I(S_{ij}; R_{ij}), \tag{6}$$

where $I(\cdot; \cdot)$ denotes mutual information.

Definition 3.2 provides an information-theoretic criterion for analyzing the effectiveness of positive alignment in GCLs. It quantifies how much information the similarity score $R_{ij}$ carries about whether a node pair is a true positive sample. Intuitively, it indicates whether positive alignment allows the encoder to distinguish true positive pairs from randomly sampled pairs based on their similarity.

Based on Lemma 3.1 and Definition 3.2, we next analyze how GNN message passing affects positive learning effectiveness. We establish this connection through Dirichlet energy. As shown in Eq. 3, the Dirichlet energy of feature $m$ at propagation layers $l$ is defined as:

$$\mathcal{E}_m^{(l)} \triangleq \left( \mathbf{H}_{:,m}^{(l)} \right)^\top \mathbf{L} \, \mathbf{H}_{:,m}^{(l)}. \tag{7}$$

Noting that the similarity score aggregates contributions from all feature dimensions, it remains unclear which features provide informative learning signals and which merely inflate similarity. We therefore decompose the similarity score into feature-wise components and analyze the contribution of each dimension separately:

$$R_{ij,m}^{(l)} \triangleq \phi_m \left( x_{i,m}^{(l)}, x_{j,m}^{(l)} \right), \qquad R_{ij}^{(l)} = \sum_{m=1}^{d} R_{ij,m}^{(l)}. \tag{8}$$

And the feature-wise positive learning effectiveness at propagation layers $l$ can be defined as follows:

$$\mathcal{I}_m^{(l)} \triangleq I \left( S_{ij}; R_{ij,m}^{(l)} \right). \tag{9}$$

We now give the following theorem:

**Theorem 3.3.** *Let $\phi$ be an inner product or cosine similarity with normalized features. Then there exists a constant $C > 0$ such that for each dimension $m$ and depth $t$,*

$$I \left( S_{ij}; R_{ij,m}^{(t)} \right) \leq C \, \mathcal{E}_m^{(t)}. \tag{10}$$

The proof can be found in Appendix E.3. Notably, under Gaussian smoothing with variance $\sigma^2$, one may take $C = \frac{B^2}{\sigma^2 \lambda_{\text{gap}}}$, where $B$ is a bound on the absolute value of each feature (e.g., $|\mathbf{x}_{i,m}^{(t)}| \leq B$), and $\lambda_{\text{gap}}$ denotes the spectral gap of the propagation operator. Theorem 3.3 shows that the contribution of each feature dimension to positive learning effectiveness is upper bounded by its Dirichlet energy, with tighter bounds for lower-energy features. We further show which feature dominate positive learning effectiveness after similarity aggregation and give the following theorem:

**Theorem 3.4.** *Let $\mathbf{R}_{ij}^{(l)} \triangleq \left( R_{ij,1}^{(l)}, \ldots, R_{ij,d}^{(l)} \right)$ be the vector of feature-wise similarity components at layer $l$, and let $\mathcal{H}$ and $\mathcal{L}$ denote the sets of high- and low-energy dimensions, respectively. Assume that $\sum_{m \in \mathcal{L}} \mathcal{E}_m^{(l)} \leq \varepsilon$. Then the following inequalities hold:*

$$I \left( S_{ij}; \mathbf{R}_{ij,\mathcal{H}}^{(l)} \right) \leq I \left( S_{ij}; \mathbf{R}_{ij}^{(l)} \right) \leq I \left( S_{ij}; \mathbf{R}_{ij,\mathcal{H}}^{(l)} \right) + C \varepsilon, \tag{11}$$

*and*

$$I \left( S_{ij}; R_{ij}^{(l)} \right) \leq I \left( S_{ij}; \mathbf{R}_{ij}^{(l)} \right), \tag{12}$$

*where $C$ is the constant in Theorem 3.3.*

Theorem 3.4 shows that aggregating similarity across all feature dimensions, low-energy dimensions can contribute at most a bounded amount of information to positive learning effectiveness, while the dominant informative signal is determined by high-energy dimensions. The proof and further implications are provided in Appendix E.4.

# 4. Methodology

Based on the theoretical analysis in Section 3, we propose **S**eparate **P**ropagation **G**raph **C**ontrastive **L**earning (SPGCL), a novel method designed to restore the learning efficacy of positive samples in graph contrastive learning. In this section, we present the overall methodology of SPGCL. An overview of the framework is shown in Figure 2. We first describe how SPGCL modifies the message passing process to prevent trivial similarity inflation and preserve learning signals that are essential for positive alignment (in Section 4.1). We then introduce a principled strategy for constructing positive samples that emphasizes stable semantic relations while reducing the influence of unreliable positives (in Section 4.2). Finally, we present the training objective that integrates these designs into a unified contrastive learning framework. Throughout this section, we use $\mathbf{Z}$ to denote the resulting node representations.

## 4.1. Energy Aware Propagation (EAP)

Given the input feature matrix $\mathbf{X}$, our goal is to identify feature dimensions that are informative for positive alignment.

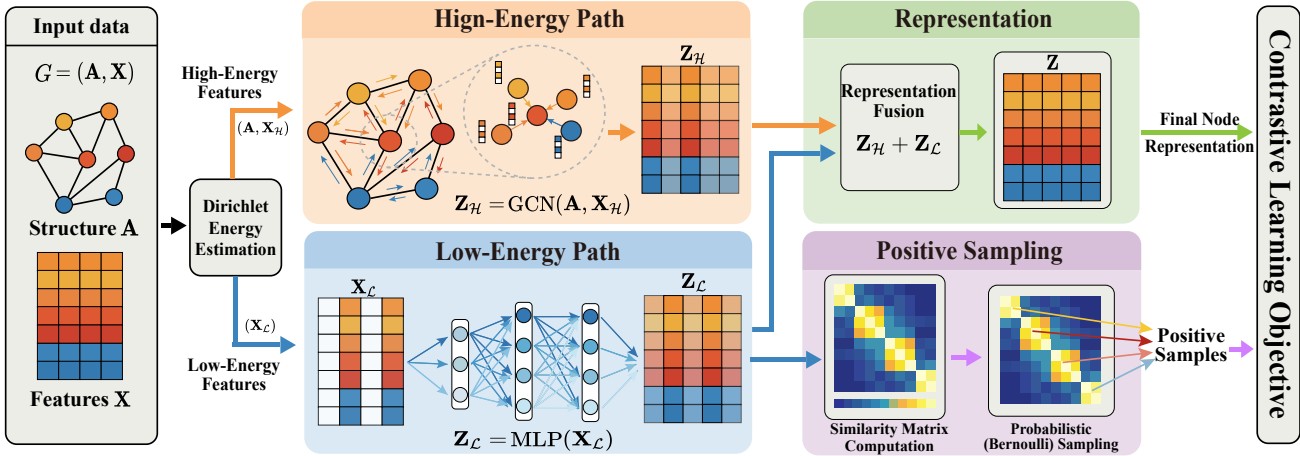

*Figure 2.* Overview of the proposed SPGCL. Given an input graph $G = (\mathbf{A}, \mathbf{X})$ with adjacency matrix $\mathbf{A}$ and node features $\mathbf{X}$, SPGCL first estimates the Dirichlet energy of each feature dimension and partitions $\mathbf{X}$ into high-energy features $\mathbf{X}_{\mathcal{H}}$ and low-energy features $\mathbf{X}_{\mathcal{L}}$. High-energy features are propagated through a GCN encoder to produce $\mathbf{Z}_{\mathcal{H}} = \text{GCN}(\mathbf{A}, \mathbf{X}_{\mathcal{H}})$, while low-energy features are transformed independently by a MLP, yielding $\mathbf{Z}_{\mathcal{L}} = \text{MLP}(\mathbf{X}_{\mathcal{L}})$. The two components are then combined via element-wise addition to obtain the final node representation $\mathbf{Z} = \mathbf{Z}_{\mathcal{H}} + \mathbf{Z}_{\mathcal{L}}$. In parallel, low-energy features are used to compute a similarity matrix. Finally, the resulting representations $\mathbf{Z}$ and the positive samples are used to optimize Eq.4.

As shown in Section 3, the contribution of each feature dimension to positive learning effectiveness is upper bounded by its Dirichlet energy. This motivates us to explicitly quantify the Dirichlet energy of individual feature dimensions and use it to guide feature partition. Following the definition in Section 2, the Dirichlet energy of the $m$-th feature dimension $\mathbf{X}_{:,m} \in \mathbb{R}^N$ can be estimated as:

$$\mathcal{E}_m = \frac{1}{2} \sum_{(v_i, v_j) \in \mathcal{T}} \mathbf{A}_{ij} (x_{i,m} - x_{j,m})^2, \qquad (13)$$

which measures the total variation of the $m$-th feature across graph edges. A larger value of $\mathcal{E}_m$ indicates that this feature exhibits stronger local variation and is therefore less affected by excessive smoothing induced by message passing. In practice, explicitly computing Eq. (13) for all feature dimensions incurs a significant computational cost. To improve efficiency, we can estimate the Dirichlet energy as follows:

$$\widehat{\mathcal{E}}_m \propto \mathbb{E}_{(v_i, v_j) \sim \mathcal{T}} \left[ (x_{i,m} - x_{j,m})^2 \right], \qquad (14)$$

where $(v_i, v_j) \sim \mathcal{T}$ denotes uniformly sampling an edge from the edge set. Since SPGCL only relies on the relative ordering of feature energies, this approximation is sufficient for guiding feature partition. Based on the estimated energies, we partition the feature dimensions into high- and low-energy subsets by selecting the TopK dimensions with the largest Dirichlet energies:

$$\mathcal{H} = \text{TopK}\left(\{\widehat{\mathcal{E}}_m\}_{m=1}^F\right), \qquad \mathcal{L} = \{1, \ldots, F\} \setminus \mathcal{H}, \quad (15)$$

where $F$ denotes the total number of feature dimensions, $\mathcal{H}$ is the set of high-energy dimensions, and $\mathcal{L}$ contains the remaining low-energy ones. This partition is computed once before training and remains fixed throughout optimization.

Following the feature partition, we separate message passing for high and low energy features to explicitly mitigate the pre-alignment effect caused by message passing. As shown in Section 3, propagating low Dirichlet energy features through message passing mainly amplifies similarity without providing effective learning signals. We aim to prevent such redundant similarity accumulation while preserving informative variations. Let $\mathbf{X}_{\mathcal{H}}$ and $\mathbf{X}_{\mathcal{L}}$ denote the feature matrices restricted to the high-energy dimension set $\mathcal{H}$ and the low-energy dimension set $\mathcal{L}$, respectively. High-energy features are propagated through a standard message passing GCN encoder:

$$\mathbf{Z}_{\mathcal{H}} = \text{GCN}(\mathbf{A}, \mathbf{X}_{\mathcal{H}}) = \sigma(\mathbf{P}\,\mathbf{X}_{\mathcal{H}}\,\mathbf{W}_{\mathcal{H}}), \qquad (16)$$

where $\text{GCN}(\cdot)$ follows the message passing formulation in Eq. 2. In contrast, low-energy features are not propagated across the graph. Instead, they are transformed independently using a feature-wise mapping:

$$\mathbf{Z}_{\mathcal{L}} = \text{MLP}(\mathbf{X}_{\mathcal{L}}) = \sigma(\mathbf{X}_{\mathcal{L}}\,\mathbf{W}_{\mathcal{L}}), \qquad (17)$$

where $\text{MLP}(\cdot)$ applies a node-wise transformation independently to each node, without involving any graph propagation. This prevents low-energy features from dominating representation similarity through excessive smoothing, thereby alleviating the pre-alignment effect identified in our theoretical analysis. We then obtain the final node representation by combining the two components via element-wise addition as follows:

$$\mathbf{Z} = \mathbf{Z}_{\mathcal{H}} + \mathbf{Z}_{\mathcal{L}}, \qquad (18)$$

where the addition is performed along the feature dimension. This representation serves as the final node embedding for training and testing.

## 4.2. Energy-guided Positive Sampling (EPS)

While low-energy features contribute little to effective positive alignment, the theoretical analysis in Section 3 shows that they encode smooth and stable signals over the graph. Such features are less sensitive to perturbations, making them suitable for identifying reliable semantic relations. We therefore use low-energy features to guide positive samples construction. Specifically, given the low-energy feature matrix $\mathbf{X}_{\mathcal{L}}$, we first perform row-wise $\ell_2$ normalization and compute a pairwise similarity matrix:

$$\mathbf{S}_{ij} = \frac{\mathbf{x}_{i,\mathcal{L}}^{\top} \mathbf{x}_{j,\mathcal{L}}}{\|\mathbf{x}_{i,\mathcal{L}}\|_2 \|\mathbf{x}_{j,\mathcal{L}}\|_2}, \tag{19}$$

where $\mathbf{x}_{i,\mathcal{L}}$ denotes the low-energy feature vector of node $v_i$. The similarity score $\mathbf{S}_{ij} \in [0, 1]$ reflects the consistency of node pairs under smooth feature components. Based on $\mathbf{S}$, we define a probabilistic positive sampling rule. For each edge $(v_i, v_j) \in \mathcal{T}$, a binary random variable is sampled as

$$\mathbf{M}_{ij} \sim \text{Bernoulli}(p_{ij}), p_{ij} = \min\left(\alpha \, \mathbf{S}_{ij}, 1\right) \tag{20}$$

where $\alpha > 0$ is a scaling hyper-parameter controlling the overall retention rate. A node pair $(v_i, v_j)$ is selected as a positive pair if $\mathbf{M}_{ij} = 1$. This sampling strategy biases positive selection toward node pairs that exhibit stable similarity under low-energy features. Such an effect is particularly beneficial on heterophilic graphs, where local neighborhoods often fail to reflect semantic consistency. The selected positive pairs are directly used in the contrastive objective. Finally, SPGCL optimizes the standard InfoNCE loss in Eq. (4). The key difference from existing GCL methods lies in how we construct the positive set $\mathcal{P}$. SPGCL does not introduce any additional loss terms. The effectiveness of SPGCL stems from the energy-aware design of feature propagation and positive sampling, which directly follows from the theoretical analysis in Section 3.

## 5. Experiments

In this section, we comprehensively evaluate the effectiveness of the proposed SPGCL by comparing it with multiple baselines on node-level tasks, including node classification and node clustering. Specifically, in Section 5.1, we describe the experimental settings, including datasets, baselines, and implementation details. In Section 5.2, we analyze the performance of SPGCL on node classification tasks under both homophilic and heterophilic graph settings, as well as on node clustering tasks. In Section 5.3, we further conduct hyper-parameter sensitivity analysis and ablation studies to quantify the contribution of each module in SPGCL.

## 5.1. Experimental Settings

**Datasets.** We evaluate SPGCL on 12 widely used datasets covering both homophilic and heterophilic graphs. For homophilic graphs, we use Cora, CiteSeer, PubMed (Sen et al., 2008; Yang et al., 2016), Photo, Computers (McAuley et al., 2015), and CS (Sinha et al., 2015). For heterophilic graphs, we use Chameleon, Cornell, Texas, Wisconsin, Crocodile and Actor (Pei et al., 2020b; Rozemberczki et al., 2021b). For detailed statistics and descriptions on these datasets, please refer to Appendix B.1.

**Baselines.** We compare SPGCL with a standard semi-supervised learning method: GCN (Kipf & Welling, 2017), and the recent state-of-the-art graph self-supervised learning methods: GRACE (Zhu et al., 2020), DGI (Velickovic et al., 2019), BGRL (Thakoor et al., 2021), SCE (Zhang et al., 2020), GCA (Zhu et al., 2021), LocalGCL (Zhang et al., 2022a), ProGCL (Xia et al., 2022), AFGRL (Lee et al., 2022), SGRL (He et al., 2024a), E2Neg (Huang et al., 2025), StrGCL (He et al., 2025), CSGCL (Chen et al., 2023), SIGNA (Sun et al., 2025), HGRL (Chen et al., 2022), DSSL (Xiao et al., 2022), GraphACL (Xiao et al., 2023). The descriptions and implementation details of baselines are given in Appendix B.2.

**Setups.** All graph self-supervised learning methods are pre-trained with a standard GCN backbone, where the number of layers follows the optimal setting reported in the original papers and their released code. We randomly initialize model parameters and train the encoder with the Adam optimizer (Kingma & Ba, 2014). Each experiment is repeated ten times, and the average results and standard deviations are reported. For node classification, we adopt a standard linear evaluation protocol. On homophilic graphs, the classifier setting follows (Thakoor et al., 2021), while on heterophilic graphs, we follow the evaluation protocol of (Xiao et al., 2023). In all cases, a linear classifier is trained on top of the frozen representations, and test accuracy is reported. For node clustering, we follow (He et al., 2024a). More detailed settings can be found in Appendix B.3.

## 5.2. Performance Analysis

**Node classification on homophilic graphs.** As shown in Table 1, SPGCL achieves the best accuracy on all six homophilic datasets. The gain mainly comes from separate propagation, which removes the trivial positive similarity introduced by message passing and makes positive pairs provide non-trivial learning signals. First, compared with methods that learn only use positive samples, such as BGRL and AFGRL, SPGCL outperforms them significantly. Although these methods avoid explicit negative sampling through bootstrap-style training strategy, they still operate on representations produced by standard GCN propagation, where message passing induces a pre-alignment effect that

*Table 1.* Node classification results on homophilic graphs. Best results are in bold, and second-best are underlined. Methods with ∗ report results from original papers. **X**, **A**, and **Y** denote node features, adjacency matrix, and node labels. GCN is drawn from (He et al., 2024b).

| Method | Training Data | Cora | CiteSeer | PubMed | Photo | CS | Computers |
|---|---|---|---|---|---|---|---|
| GCN | **X, A, Y** | 82.80 ± 0.00 | 72.00 ± 0.00 | 84.80 ± 0.00 | 93.03 ± 0.00 | 93.03 ± 0.00 | 86.51 ± 0.00 |
| GRACE | **X, A** | 84.23 ± 0.37 | 70.77 ± 0.50 | 86.02 ± 0.19 | 92.57 ± 0.34 | 93.05 ± 0.10 | 88.78 ± 0.15 |
| DGI | **X, A** | 83.72 ± 0.55 | 70.52 ± 0.64 | 86.12 ± 0.22 | 92.94 ± 0.04 | 93.17 ± 0.11 | 87.89 ± 0.45 |
| BGRL | **X, A** | 83.25 ± 0.82 | 70.32 ± 0.77 | 85.57 ± 0.34 | 93.26 ± 0.16 | 93.41 ± 0.11 | 89.41 ± 0.09 |
| SCE | **X, A** | 84.72 ± 0.70 | 72.67 ± 0.70 | 85.03 ± 0.25 | 92.67 ± 0.13 | 93.06 ± 0.20 | 89.45 ± 0.27 |
| GCA | **X, A** | 85.08 ± 0.47 | 71.22 ± 0.49 | 86.46 ± 0.22 | 92.84 ± 0.26 | 93.48 ± 0.09 | 89.95 ± 0.02 |
| LocalGCL | **X, A** | 84.82 ± 0.79 | 69.48 ± 0.35 | 86.53 ± 0.25 | 93.56 ± 0.17 | 92.52 ± 0.13 | 89.59 ± 0.13 |
| ProGCL | **X, A** | 84.91 ± 0.85 | 71.73 ± 0.26 | 86.97 ± 0.18 | 93.09 ± 0.24 | 93.36 ± 0.09 | 88.09 ± 0.11 |
| AFGRL* | **X, A** | - | - | - | 93.22 ± 0.28 | 93.27 ± 0.17 | 89.88 ± 0.33 |
| SGRL | **X, A** | 83.93 ± 0.30 | 70.41 ± 0.62 | 86.02 ± 0.25 | 93.95 ± 0.01 | 94.08 ± 0.08 | 90.03 ± 0.05 |
| E2Neg | **X, A** | 84.20 ± 0.36 | 70.30 ± 0.61 | 87.08 ± 0.12 | 93.10 ± 0.26 | 92.99 ± 0.06 | 89.02 ± 0.36 |
| StrGCL | **X, A** | 84.86 ± 0.54 | 72.49 ± 0.11 | 86.52 ± 0.19 | 93.94 ± 0.12 | 94.06 ± 0.03 | 90.32 ± 0.05 |
| CSGCL | **X, A** | 82.66 ± 0.43 | 69.91 ± 0.47 | 86.48 ± 0.24 | 93.21 ± 0.07 | 93.63 ± 0.13 | 90.67 ± 0.21 |
| SIGNA | **X, A** | 82.19 ± 0.35 | 69.41 ± 0.47 | 84.23 ± 0.09 | 93.57 ± 0.05 | 93.12 ± 0.05 | 90.90 ± 0.09 |
| **SPGCL(Ours)** | **X, A** | **85.44 ± 0.13** | **73.49 ± 0.26** | **87.19 ± 0.13** | **94.83 ± 0.07** | **95.03 ± 0.03** | **91.04 ± 0.09** |

*Table 2.* Node clustering performance evaluated by NMI and Homogeneity. Best results are in bold and second-best are underlined.

| | | GRACE | DGI | BGRL | SGRL | **SPGCL** |
|---|---|---|---|---|---|---|
| Computers | NMI | 0.4793 | 0.4630 | 0.5364 | 0.5380 | **0.5626** |
| | Hom. | 0.5222 | 0.4836 | 0.5869 | 0.5705 | **0.6132** |
| Photo | NMI | 0.6513 | 0.5487 | 0.6841 | 0.6788 | **0.7135** |
| | Hom. | 0.6657 | 0.5557 | 0.7004 | 0.6786 | **0.7251** |
| CS | NMI | 0.7562 | 0.7162 | 0.7732 | 0.7961 | **0.8046** |
| | Hom. | 0.7909 | 0.7428 | 0.8041 | 0.8216 | **0.8345** |

trivializes positive similarity. As a result, the learning signal extracted from positive pairs remains limited, whereas SPGCL explicitly mitigates this issue through energy-aware feature propagation. Moreover, SPGCL outperforms methods that improve data augmentation or positive sampling strategies, including GCA, LocalGCL, and CSGCL. These methods strengthen positive construction through improved perturbations or sampling rules, but they still build positives on embeddings that have already been partly aligned by message passing. This limits the effective learning signals from positive samples. Finally, methods that focus on negative sample optimization, such as ProGCL and SGRL, achieve competitive but consistently inferior performance compared to SPGCL. This suggests that refining negative samples alone is insufficient to exploit the full potential of contrastive learning, and that effectively restoring informative positive learning plays a more critical role in representation quality.

**Node classification on heterophilic datasets.** To further verify the effectiveness of SPGCL, we evaluate it on six widely used heterophilic graphs, and the results are reported in Table 3. Compared with methods designed for homophilic graphs, including GRACE, DGI,

BGRL, GCA, and LocalGCL, SPGCL achieves significant performance improvements across all datasets. This gap mainly arises from the positive sampling strategies (augmentation-based or neighbor-based) adopted by these methods. Augmentation-based methods construct positive samples by perturbing node features or graph structures, which destroy core semantic information in heterophilic graphs, leading to unreliable positives. Neighbor-based positive sampling, which works well under the homophily assumption, becomes ineffective in heterophilic settings, as adjacent nodes often belong to different classes and enforcing their alignment introduces misleading signals. Moreover, SPGCL also outperforms methods designed for heterophilic graphs. Although these methods introduce different designs to handle heterophily, such as modeling higher-order dependencies (Chen et al., 2022), latent semantics (Xiao et al., 2022), or asymmetric neighborhoods (Xiao et al., 2023), they all operate on representations where all feature dimensions are uniformly propagated through the graph. As a result, positive samples are aligned without distinguishing which feature dimensions contribute to their similarity. In contrast, SPGCL explicitly separates feature propagation based on Dirichlet energy, which introduces feature-level control over how positive similarity is constructed.

**Node clustering.** We follow the evaluation protocol and report the results from (He et al., 2024a). As shown in Table 2, SPGCL outperforms all baselines. Notably, SPGCL shows a pronounced improvement on Photo, surpassing the second-best method by 2.94% in NMI and 2.47% in Homogeneity, indicating substantially better cluster separation. These results indicate that SPGCL produces representations where nodes with the same labels are more likely to be grouped into the same clusters. This stems from more effec-

*Table 3.* Node classification results on heterophilic graphs. Best results are in bold, and second-best are underlined. Methods with ∗ report results from (Xiao et al., 2023). **X**, **A**, and **Y** denote node features, adjacency matrix, and node labels.

| Method | Training Data | Chameleon | Cornell | Texas | Wisconsin | Crocodile | Actor |
|---|---|---|---|---|---|---|---|
| GCN | **X, A, Y** | $62.15 \pm 2.47$ | $43.51 \pm 8.87$ | $58.73 \pm 4.77$ | $56.86 \pm 7.95$ | $64.31 \pm 0.71$ | $26.43 \pm 1.21$ |
| MLP | **X, Y** | $49.06 \pm 1.86$ | $50.91 \pm 8.99$ | $58.92 \pm 4.56$ | $65.49 \pm 4.55$ | $64.41 \pm 0.53$ | $29.88 \pm 1.12$ |
| GRACE | **X, A** | $64.58 \pm 1.84$ | $42.70 \pm 7.62$ | $60.27 \pm 5.10$ | $58.04 \pm 7.05$ | $71.32 \pm 0.33$ | $29.93 \pm 0.91$ |
| DGI | **X, A** | $64.91 \pm 1.86$ | $50.00 \pm 6.65$ | $63.78 \pm 4.45$ | $56.86 \pm 6.98$ | $71.26 \pm 0.59$ | $29.28 \pm 1.11$ |
| BGRL | **X, A** | $63.86 \pm 2.89$ | $43.70 \pm 5.81$ | $59.19 \pm 6.91$ | $52.16 \pm 6.21$ | $54.35 \pm 0.89$ | $24.83 \pm 0.99$ |
| GCA | **X, A** | $60.35 \pm 2.69$ | $47.57 \pm 6.88$ | $63.24 \pm 4.80$ | $57.84 \pm 8.48$ | $68.87 \pm 0.88$ | $29.70 \pm 1.04$ |
| LocalGCL | **X, A** | $61.86 \pm 1.77$ | $37.30 \pm 9.08$ | $52.70 \pm 5.73$ | $46.47 \pm 5.47$ | $63.04 \pm 0.48$ | $25.37 \pm 0.49$ |
| HGRL* | **X, A** | $65.82 \pm 0.61$ | $51.78 \pm 1.03$ | $61.83 \pm 0.71$ | $63.90 \pm 0.58$ | $61.87 \pm 0.45$ | $27.95 \pm 0.30$ |
| DSSL* | **X, A** | $66.15 \pm 0.32$ | $53.15 \pm 1.28$ | $62.11 \pm 1.53$ | $62.25 \pm 0.55$ | $62.98 \pm 0.51$ | $28.15 \pm 0.31$ |
| GraphACL | **X, A** | $\underline{69.12 \pm 2.11}$ | $\underline{60.54 \pm 3.86}$ | $\underline{71.62 \pm 7.12}$ | $\underline{70.20 \pm 5.61}$ | $\underline{66.09 \pm 1.00}$ | $\underline{29.95 \pm 0.93}$ |
| **SPGCL(Ours)** | **X, A** | $\mathbf{72.26 \pm 1.66}$ | $\mathbf{75.41 \pm 6.43}$ | $\mathbf{80.81 \pm 6.30}$ | $\mathbf{83.53 \pm 4.26}$ | $\mathbf{77.43 \pm 0.67}$ | $\mathbf{37.23 \pm 1.19}$ |

*Table 4.* Ablation study on node classification task.

| Variant | Cora | Photo | CS | Cornell | Texas | Actor |
|---|---|---|---|---|---|---|
| **SPGCL** | $\mathbf{85.44 \pm 0.13}$ | $\mathbf{94.83 \pm 0.07}$ | $\mathbf{95.03 \pm 0.03}$ | $\mathbf{75.41 \pm 6.43}$ | $\mathbf{80.81 \pm 6.30}$ | $\mathbf{37.23 \pm 1.19}$ |
| w/o EAP | $84.10 \pm 0.21$ | $93.21 \pm 0.12$ | $92.99 \pm 0.15$ | $50.27 \pm 5.87$ | $68.91 \pm 8.76$ | $24.86 \pm 1.20$ |
| w/o EPS | $84.91 \pm 0.38$ | $93.99 \pm 0.11$ | $94.67 \pm 0.01$ | $66.76 \pm 5.41$ | $78.65 \pm 7.48$ | $35.00 \pm 1.03$ |
| w/o EAP & EPS | $83.70 \pm 0.24$ | $93.73 \pm 0.08$ | $93.67 \pm 0.04$ | $50.08 \pm 7.48$ | $66.76 \pm 7.76$ | $23.33 \pm 1.27$ |

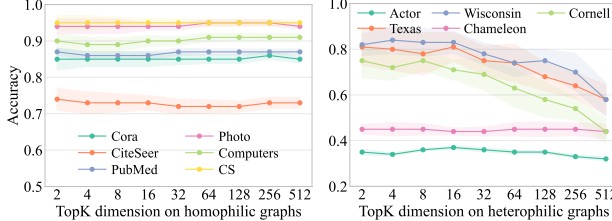

*Figure 3.* Hyper-parameter analysis of SPGCL with respect to the TopK threshold. We report the classification performance on both homophilic (left) and heterophilic (right) graphs.

tive positive learning in SPGCL, leading to higher-quality representations. We further visualize the learned embeddings using t-SNE, and present the results in the Appendix.

### 5.3. Model Analysis

Table 4 reports the ablation results on homophilic and heterophilic datasets. Removing either Energy-Aware Propagation (EAP) or Energy-guided Positive Sampling (EPS) consistently degrades performance, and the drop is particularly significant when EAP is removed. This confirms that separating feature propagation based on Dirichlet energy is the core factor for restoring effective positive learning. We also conduct hyper-parameter experiments. Figure 3 analyzes the impact of the TopK used to select high-energy feature dimensions. On homophilic graphs, performance remains

relatively stable as TopK increases, with a slight downward trend. This reflects a trade-off between two effects: expanding the high-energy subset weakens the selectivity of feature propagation for positive alignment, while simultaneously enlarging the low-energy subset used for positive sampling, which yields more stable positive pairs. These two effects partially offset each other, resulting in overall stable performance. In contrast, on heterophilic graphs, performance exhibits a clear decreasing trend as TopK increases. In this setting, propagating a broader set of features amplifies the mixing of semantically dissimilar neighbors, causing positive pairs from different classes to become overly similar. Although low-energy features can still identify relatively reliable positives, the degradation of positive learning signals induced by propagation dominates, leading to a more pronounced performance drop.

### 6. Conclusion

In this paper, we reveal that positive sampling may not work in GCL as be widely recognized. Empirically, we observe that existing GCL methods can achieve competitive performance without positive samples. Theoretically we attribute this phenomenon to the pre-alignment effect, where the message passing of graph neural network increases the similarity between positive samples before contrastive optimization. We further show that the learning effectiveness of positive samples is upper bounded by feature-wise Dirichlet energy. Pre-alignment reduces the Dirichlet energy, thus weakens

learning from positive samples. To address this, we propose SPGCL, which separates feature propagation according to Dirichlet energy: high-energy features are propagated to preserve informative variation for positive learning, while low-energy features are exploited to guide reliable positive sampling. Extensive experiments demonstrate that SPGCL can effectively learn from positive samples and outperforms existing GCL methods.

## Acknowledgments

This work was supported by the National Natural Science Foundation of China (No. 62276187, No. 62422210, No.92370111, and No. 62272340).

## Impact Statement

This paper aims to contribute to the theoretical understanding of graph contrastive learning within the broader field of machine learning. By analyzing how message passing influences learning from positive samples, our work offers a perspective on contrastive pre-training paradigms, which is widely used in graph representation learning. We hope that this study may help clarify certain aspects of positive sample utilization and support future investigations into contrastive objectives for graph models.

This work is primarily theoretical in nature and does not introduce new datasets, deployable systems, or models intended for real-world use. As such, we do not anticipate any direct or indirect negative societal impacts or ethical concerns arising from this study. The proposed analysis does not involve sensitive or personal data, human subjects, or crowdsourcing, nor does it introduce methods that could be readily misused for harmful purposes. To the best of our knowledge, this research does not raise ethical issues.

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

# A. Related Work

**Graph Representation Learning.** Graph Representation Learning (GRL) aims to embed high-dimensional, sparse, and non-Euclidean graph-structured data into low-dimensional representations for various downstream tasks, such as node classification, clustering, and link prediction (Chen et al., 2020). Early GRL methods primarily rely on matrix factorization (Zhang, 2022) or random walk–based techniques (Lawler & Limic, 2010), which focus on either graph structure or node attributes and thus struggle to jointly model both (Ribeiro et al., 2017). With the emergence of Graph Neural Networks (GNNs), such as GCN (Kipf & Welling, 2017), GAT (Velickovic et al., 2018), and GraphSAGE (Hamilton et al., 2017), message passing and aggregation mechanism makes it possible to jointly embed graph structure and node attributes in representations. Despite their success, GNNs typically require large amounts of labeled data for supervision, which is often expensive and impractical in real-world (Liu et al., 2022b). To address this limitation, Graph Contrastive Learning (GCL) has emerged as a mainstream self-supervised pre-training paradigm, which trains graph encoders by maximizing the similarity between positive samples and minimizing it between negative samples without labels.

**Graph Contrastive Learning.** In GCLs, it is a consensus that positive samples play a key role, as contrastive objectives rely on aligning semantically consistent views of the same node or subgraph to learn invariant representations (Zhu et al., 2020; Velickovic et al., 2019; Thakoor et al., 2021; Liu et al., 2022a; Zhuo et al., 2024a). GCLs typically construct positive samples in two ways: (i) the same anchor node across multiple views generated by data augmentations, and (ii) its local neighbors sampled from graphs. Consequently, many existing methods focus on refining positive samples through improved data augmentation and neighborhood-based positive sampling. Early GCLs adopt random feature masking and edge dropping to generate positive views (Zhu et al., 2020; Velickovic et al., 2019; Thakoor et al., 2021; Zhuo et al., 2024b). However, such random perturbations may destroy critical structural and semantic information, leading to misleading positive pairs. To address this, some methods design augmentation strategies based on graph properties. GCA (Zhu et al., 2021) and GRADE (Wang et al., 2022) improve data augmentation according to node centrality or degree, reducing the risk that positive samples become semantically inconsistent due to excessive structural corruption. CSGCL (Chen et al., 2023) incorporates community structure into augmentation design, encouraging positive samples to preserve consistent community-level semantics. COSTA (Zhang et al., 2022b) constrains feature perturbations by preserving feature covariance, so that positive samples constructed from different views retain consistent attribute semantics. Moreover, some methods improve data augmentation from a spectral perspective to enhance the quality of positive samples. SpCo (Liu et al., 2022a) reveals that effective augmentations should preserve low-frequency components while perturbing high-frequency ones. It perturbs edges with high-frequency while retaining low-frequency edges during augmentations. However, these methods rely on manually designed rules, which limits their adaptability across diverse graphs. To address this, AD-GCL (Suresh et al., 2021) trains a neural network to generate diverse views by minimizing mutual information between original and augmented graphs, improving the construction of informative positive samples without predefined heuristics augmentation. Notably, BGRL (Thakoor et al., 2021) and AFGRL (Lee et al., 2022) adopt a bootstrap framework that trains models without negative samples. They employ two asymmetric graph encoders and learn representations by aligning positive pairs across different views. These methods represent an early attempt to exploit positive samples learning in GCLs. However, recent studies show that the effectiveness of such bootstrap frameworks is highly sensitive to batch normalization, which implicitly introduces negative sample information (Fetterman & Albrecht, 2020; He et al., 2024a). Consequently, the utilization of positive samples in these methods remains limited.

In contrast to prior work that mainly focuses on constructing better positive samples, we revisit how positive samples are actually utilized in GCLs. We show that message passing induces a pre-alignment effect, which increases positive similarity before contrastive optimization and consequently weakens the effective learning signal from positive samples. This perspective departs from existing augmentation and sampling strategies and motivates a novel design to restore informative positive learning in GCLs.

# B. Experimental Settings

### B.1. Datasets

In this subsection, we describe the datasets used for experimental evaluation in detail. We evaluate SPGCL on diverse graph datasets with substantially different structural characteristics, providing a comprehensive assessment on node-level tasks. Detailed statistics of each dataset are summarized in Table 5.

**Citation networks.** Cora, CiteSeer, and PubMed are widely used citation datasets from Planetoid (Sen et al., 2008; Yang

et al., 2016). Each node denotes a paper and each edge denotes a citation relation. Node attributes come from sparse bag-of-words representations of paper text (e.g., titles and abstracts). The class labels indicate the research area of paper, such as machine learning or neural networks.

**Co-purchase networks.** Photo and Computers (McAuley et al., 2015) describe co-purchasing relations in e-commerce. Nodes are products in their respective catalogs. Edges link items that are often purchased together. Each node uses sparse text features extracted from user reviews. Labels mark high-level product types..

**Co-author networks.** The Co.CS dataset is built from the Microsoft Academic Graph (MAG) (Sinha et al., 2015). Nodes represent authors, and edges indicate co-authored papers. Node features use keyword-based statistics from an author's publications. Labels reflect the author's main research area inferred from their publication records.

**WebKB.** WebKB[2] contains web pages from computer science departments at several universities, originally collected at Carnegie Mellon University. Cornell, Texas, and Wisconsin are used in our experiments. Nodes correspond to pages and edges correspond to hyperlinks. Each page is represented with sparse word features extracted from its content. Labels are human-annotated into five classes: student, project, course, staff, and faculty.

**Actor co-occurrence network.** This graph comes from an actor-only subgraph of a larger film-related network (Tang et al., 2009). Each node denotes an actor. An edge indicates that two actors appear together on the same Wikipedia page. Node attributes are sparse keyword features extracted from the page text. Actors are grouped into five classes based on the content of their Wikipedia pages.

**Wikipedia network.** Chameleon and Squirrel are page graphs built from Wikipedia (Rozemberczki et al., 2021a). Nodes correspond to pages, and edges indicate links between pages. Each page uses sparse keyword features extracted from its text. Labels form five classes based on the page's average monthly traffic.

We obtain these datasets from the public PyTorch Geometric (PyG) repository (Fey & Lenssen, 2019). All datasets can be accessed through URLs listed below:

- Cora, CiteSeer, PubMed: `https://github.com/kimiyoung/planetoid/raw/master/data`.

- Photo, Computers, CS: `https://github.com/shchur/gnn-benchmark/raw/master/data/npz`.

- Crocodile: `https://graphmining.ai/datasets/ptg/wiki`.

- Chameleon, Actor, Cornell, Texas, Wisconsin: `https://github.com/bingzhewei/geom-gcn/tree/master/splits`.

*Table 5.* Dataset statistics.

| Datasets | Cora | CiteSeer | PubMed | Photo | Computers | CS | Actor | Chameleon | Crocodile | Cornell | Texas | Wisconsin |
|---|---|---|---|---|---|---|---|---|---|---|---|---|
| **Nodes** | 2,708 | 3,327 | 19,717 | 7,650 | 13,752 | 18,333 | 7,600 | 2,277 | 11,631 | 183 | 183 | 251 |
| **Edges** | 4,732 | 5,429 | 44,338 | 119,081 | 245,861 | 81,894 | 33,544 | 36,101 | 170,918 | 295 | 309 | 499 |
| **Features** | 1,433 | 3,703 | 500 | 745 | 767 | 6,805 | 932 | 2,325 | 128 | 1,703 | 1,703 | 1,703 |
| **Classes** | 7 | 6 | 3 | 8 | 10 | 15 | 5 | 5 | 7 | 5 | 5 | 5 |

### B.2. Introductions of Baselines.

**GCN** (Kipf & Welling, 2017): This method performs layer-wise feature propagation using a first-order approximation of spectral graph convolutions, enabling nodes to aggregate information from their neighbors through normalized adjacency and node features for semi-supervised node classification.

**GRACE** (Zhu et al., 2020): This method generates two augmented views by applying random perturbations to node features and edges, and trains graph encoders with the InfoNCE loss by maximizing the similarity between positive samples while minimizing the similarity to negative samples.

**DGI** (Velickovic et al., 2019): This method generates a corrupted graph by randomly shuffling node features and trains a mutual-information discriminator to distinguish node embeddings from the original graph against those from the corrupted

---

[2]`http://www.cs.cmu.edu/afs/cs.cmu.edu/project/theo-11/www/wwkb`

one. The objective maximizes the mutual information between node representations learned on the original graph and a global summary vector of the same graph, enabling effective representation learning without labels.

**BGRL** (Thakoor et al., 2021): This method adopts a bootstrap framework to train a graph encoder without negative samples. Similar to GRACE, two augmented views are generated through stochastic perturbations of node features and graph structure, and the training objective enforces consistency between the representations of corresponding nodes across these views. The architecture includes an online encoder with a predictor and a momentum-updated target network.

**GCA** (Zhu et al., 2021): GCA: This method retains GRACE-style node-level contrastive learning, while replacing random augmentations with adaptive edge dropping and feature masking guided by node centrality. The augmentation preserves structurally important components more conservatively and applies stronger corruption to less informative regions, balancing structure preservation with effective perturbations for representation learning.

**LocalGCL** (Zhang et al., 2022a): This method performs node-level contrastive learning without graph augmentations and defines one-hop neighbors as positive samples. The contrastive objective takes a kernelized form, and random Fourier features approximate the kernel to enable scalable loss computation.

**ProGCL** (Xia et al., 2022): This method provides a hard-negative mining framework for node-level graph contrastive learning, with explicit exploiting "false hard negatives" in GCLs. It models each negative sample with a beta mixture model to estimate the probability of being a true negative, and incorporates this estimate into negative reweighting or negative mixing to strengthen the contrastive learning.

**AFGRL** (Lee et al., 2022): This method also adopts bootstrap framework. It is an augmentation-free graph contrastive learning method that constructs positive samples through k-NN retrieval in the embedding space and refines them with both local adjacency constraints and global clustering signals. It aligns the online encoder with a momentum target encoder on the selected positives, where the online network predicts the target representations.

**SGRL** (He et al., 2024a): This method proposes representation scattering mechanism to train graph encoders. It learns embeddings with an explicit center-away scattering loss and a topology-based constraint to keep structure-aware diversity without manual negatives.

**CSGCL** (Chen et al., 2023): This method incorporates community strength into both view generation and model training. It introduces strength-aware attribute voting and community-guided edge dropping to construct informative augmented views. Training follows a two-stage Team-up objective that progressively injects community-strength bias into InfoNCE-based similarity estimation, enabling more structure-aware contrastive learning.

**SIGNA** (Sun et al., 2025): This method is a single-view graph contrastive learning method that introduces dropout noise in embeddings and applies stochastic neighbor masking to select only a subset of neighbors as positive samples. It optimizes a normalized Jensen–Shannon divergence objective, providing a soft, neighborhood-aware training signal without cross-view augmentations.

**HGRL** (Chen et al., 2022): This method is a self-supervised framework designed for heterophilic graphs. It learns node representations through two mutual-information-based pretext tasks: one preserves original node features, and the other captures informative signals from distant neighbors. These objectives are integrated within a multi-view learning setup to model heterophily-aware dependencies.

**DSSL** (Xiao et al., 2022): This method is a variational latent-variable self-supervised framework that decouples heterogeneous neighborhood semantics through a mixture-based neighbor generation model. The latent mixture components capture diverse neighbor relations, supporting augmentation-free representation learning on heterophilic graphs.

**GraphACL** (Xiao et al., 2023): This method designs an augmentation-free asymmetric framework. An online encoder with a predictor is trained against an EMA teacher to predict one-hop neighbor context, capturing one-hop contextual signals and inducing two-hop "monophily" effects that benefit representation learning on heterophilic graphs.

We implement GCN via the PyTorch Geometric (PyG) library. For self-supervised baselines, we employ an official implementation publicly released by the original papers. The corresponding sources are listed as follows:

- GCN: `https://github.com/pyg-team/pytorch_geometric/tree/master/torch_geometric/nn/conv`.

- GRACE: `https://github.com/CRIPAC-DIG/GRACE`.

- DGI: https://github.com/PetarV-/DGI.

- BGRL: https://github.com/nerdslab/bgrl.

- GCA: https://github.com/CRIPAC-DIG/GCA/.

- LocalGCL:
  https://openreview.net/attachment?id=dSYkYNNZkV&name=supplementary_material.

- ProGCL: https://github.com/junxia97/ProGCL.

- AFGRL: https://github.com/Namkyeong/AFGRL.

- SGRL: https://github.com/hedongxiao-tju/SGRL.

- CSGCL: https://github.com/HanChen-HUST/CSGCL.

- SIGNA: https://github.com/sunisfighting/SIGNA.

- HGRL: https://github.com/yifanQi98/HGRL.

- DSSL: https://papers.nips.cc/paper_files/paper/2022/file/
  040c816286b3844fd78f2124eec75f2e-Supplemental-Conference.zip.

- GraphACL: https://github.com/tengxiao1/GraphACL.

### B.3. Experimental Details.

**Dataset splitting.** For the homophilic datasets (Cora, CiteSeer, PubMed, Photo, Coauthor-CS, and Computers), we randomly split the nodes into training, validation and test sets with a ratio of 1:1:8. For the heterophilic datasets (Texas, Wisconsin, Cornell, Chameleon, Crocodile, and Actor), we follow Geom-GCN (Pei et al., 2020a) and use the provided raw data with the standard fixed 10-fold splits.

**Experiment Setup.** All graph self-supervised learning baselines are pre-trained with a GCN backbone encoder, using one or two layers following the settings reported in the original papers when available. For methods that do not specify the number of layers, we evaluate both one-layer and two-layer GCNs and report the best results. After pre-training, the learned node embeddings are used for downstream evaluation, including node classification and node clustering. For classification, we follow the standard linear evaluation protocol of BGRL (Thakoor et al., 2021). We train an $\ell_2$-regularized logistic regression classifier on top of the frozen embeddings. We report the mean and standard deviation of the F1 score. For node clustering, obtained representations are fed into a randomly initialized K-Means (McQueen, 1967) predictor. We set the cluster number to the ground-truth class count and report the best Normalized Mutual Information (NMI) and Homogeneity Score (Hom).

**Environment.** All experiments were run on two Linux servers, as summarized in Table 6.

*Table 6.* Experimental environment servers.

|  | Server 1 | Server 2 |
|---|---|---|
| OS | Linux 6.8.0-87-generic | Linux 6.14.0-33-generic |
| CPU | Intel(R) Xeon(R) Silver 4410Y | Intel(R) Core(TM) i5-12400 CPU @ 3.00GHz |
| GPU | Nvidia GeForce RTX 5090 | Nvidia GeForce RTX 3090 |

*Table 7.* Hyper-parameters of SPGCL on node classification task.

|  | $\alpha$ | TopK | $\tau$ | Hidden dim. | Projection dim. | Learning rate | Epochs | Dropout | Num layers |
|---|---|---|---|---|---|---|---|---|---|
| Cora | 0.8 | 256 | 0.8 | 256 | 512 | 0.0005 | 1000 | 0.3 | 2 |
| CiteSeer | 0.8 | 768 | 0.8 | 1024 | 256 | 0.0001 | 300 | 0.4 | 2 |
| PubMed | 0.8 | 256 | 0.5 | 1024 | 256 | 0.001 | 1000 | 0.2 | 1 |
| Photo | 0.2 | 256 | 0.5 | 1024 | 256 | 0.001 | 1000 | 0.1 | 2 |
| CS | 0.8 | 32 | 0.5 | 512 | 256 | 0.0001 | 1000 | 0.4 | 1 |
| Computers | 0.8 | 512 | 0.5 | 1024 | 512 | 0.0005 | 1000 | 0.4 | 2 |
| Actor | 0.4 | 8 | 1 | 4096 | 1024 | 0.001 | 50 | 0.1 | 1 |
| Texas | 0.2 | 16 | 0.75 | 2048 | 1024 | 0.005 | 40 | 0.06 | 1 |
| Wisconsin | 0.4 | 4 | 0.5 | 2048 | 1024 | 0.005 | 50 | 0.04 | 1 |
| Cornell | 0.8 | 2 | 0.5 | 2048 | 1024 | 0.005 | 50 | 0.08 | 1 |
| Crocodile | 0.4 | 1024 | 0.5 | 4096 | 1024 | 0.0005 | 60 | 0.1 | 1 |

*Table 8.* Hyper-parameters of SPGCL on node clustering task.

|  | $\alpha$ | TopK | $\tau$ | Hidden dim | Proj dim | Learning rate | Epochs | Dropout | Num layers |
|---|---|---|---|---|---|---|---|---|---|
| Photo | 0.1 | 256 | 0.5 | 256 | 1024 | 0.0005 | 1000 | 0.3 | 2 |
| CS | 0.8 | 128 | 0.5 | 256 | 256 | 0.0001 | 1000 | 0.4 | 1 |
| Computers | 0.8 | 256 | 0.5 | 512 | 1024 | 0.0005 | 1000 | 0.4 | 2 |

## C. Algorithm Description.

---
**Algorithm 1** SPGCL
---
**Input:** Graph $\mathcal{G} = (\mathcal{V}, \mathcal{T})$, hyper-parameters TopK and $\alpha$
**Output:** Node representations $\mathbf{Z}$
   **for** $m$ in 1 to $F$ **do**
      Compute Dirichlet energies $\widehat{\mathcal{E}}_m$ as in Equation (13);
   **end for**
   Select the TopK dimensions $\mathcal{H}$ with $\{\widehat{\mathcal{E}}_m\}_{m=1}^F$;
   Compute pairwise similarity matrix $\mathbf{S}$ as in Equation (19);
   Construct positive-pair sampling matrix $\mathbf{M}$ with $\mathbf{S}$ and $\alpha$ as in Equation (20);
   Construct positive pair set $\mathcal{P}$ with $\mathbf{M}$;
   Construct negative pair set $\mathcal{N}_i$ with other nodes;
   **while** not converged **do**
      Obtain embeddings $\mathbf{Z}_\mathcal{H} \leftarrow \text{GCN}(\mathbf{X}_\mathcal{H}, \mathbf{A})$ and $\mathbf{Z}_\mathcal{L} \leftarrow \text{MLP}(\mathbf{X}_\mathcal{L})$;
      Compute $\mathcal{L} \leftarrow \mathcal{L}_{\text{InfoNCE}}$ via Equation (4);
      Do backpropagation with $\mathcal{L}$;
   **end while**
   **return** $\mathbf{Z} = \mathbf{Z}_\mathcal{H} + \mathbf{Z}_\mathcal{L}$;
---

## D. Hyper-parameter Settings

In hyper-parameter search, we adjust the value of $\alpha$ and TopK in SPGCL, as well as other hyper-parameters, including temperature parameter $\tau$, hidden dimension, projection dim, learning rate and number of layers. We apply the grid search strategy to choose the optimal hyper-parameters. Specifically, we search TopK in $\{2, 4, 8, 16, 32, 64, 128, 256, 512, 1024\}$, $\alpha$ in $\{0.2, 0.4, 0.6, 0.8\}$, hidden dim and proj dim in $\{256, 512, 1024\}$, learning rate in $\{0.0001, 0.0005, 0.001, 0.005, 0.01\}$, number of layers in $\{1, 2\}$. Additionally, to mitigate overfitting during training, we follow (Sun et al., 2025) and apply Dropout (Srivastava et al., 2014) to the GCN encoder. The dropout is selected from $\{0.1, 0.2, 0.3, 0.4, 0.5\}$. Tables 7 and 8 provide the hyper-parameter settings for SPGCL on node classification and clustering tasks.

# E. Theoretical analysis

### E.1. Proof for Lemma 3.1

We provide proofs of Lemma 3.1 under two commonly used positive construction rules in graph contrastive learning: (i) augmentation-based positive samples constructed from two augmented views of the same anchor node, and (ii) neighborhood-based positive samples constructed by sampling one-hop neighbors. We consider cosine similarity with $\ell_2$-normalized node representations, i.e., $R_{ij}^{(l)} = \left\langle \bar{\mathbf{H}}_{i,:}^{(l)}, \bar{\mathbf{H}}_{j,:}^{(l)} \right\rangle$, where $\bar{\mathbf{H}}_{i,:}^{(l)} \triangleq \mathbf{H}_{i,:}^{(l)} / \|\mathbf{H}_{i,:}^{(l)}\|_2$.

**Augmentation-based positive samples.** We consider two augmented views of the same graph, indexed by $(a)$ and $(b)$. Each view applies stochastic perturbations to both node features and edges, producing $(\mathbf{X}^{(a)}, \mathbf{A}^{(a)})$ and $(\mathbf{X}^{(b)}, \mathbf{A}^{(b)})$. Let $\mathbf{P}^{(a)}$ and $\mathbf{P}^{(b)}$ be the corresponding propagation operators constructed from $\mathbf{A}^{(a)}$ and $\mathbf{A}^{(b)}$. The message passing in view $\mu \in \{a, b\}$ is:

$$\mathbf{H}^{(0,\mu)} = \mathbf{X}^{(\mu)}, \qquad \mathbf{H}^{(l+1,\mu)} = \mathbf{P}^{(\mu)} \mathbf{H}^{(l,\mu)}.$$

A positive pair is constructed by taking the same anchor node $v_i$ from the two views. We assume the two perturbations are independent and unbiased: $\mathbb{E}[\mathbf{X}^{(a)}] = \mathbb{E}[\mathbf{X}^{(b)}] = \mathbf{X}$, and similarly for edge perturbations. Define the layer-$l$ positive similarity

$$R_{i,i}^{(l)} \triangleq \left\langle \bar{\mathbf{H}}_{i,:}^{(l,a)}, \bar{\mathbf{H}}_{i,:}^{(l,b)} \right\rangle.$$

*Proof.* For unit-norm vectors, cosine similarity can be equivalently written in terms of squared Euclidean distance:

$$\langle \mathbf{u}, \mathbf{v} \rangle = 1 - \tfrac{1}{2}\|\mathbf{u} - \mathbf{v}\|_2^2, \qquad \|\mathbf{u}\|_2 = \|\mathbf{v}\|_2 = 1.$$

Therefore, proving $\mathbb{E}[R_{i,i}^{(l+1)}] \geq \mathbb{E}[R_{i,i}^{(l)}]$ is equivalent to showing that the expected squared distance between the normalized representations of the two views does not increase with depth, i.e.,

$$\mathbb{E}\left[\|\bar{\mathbf{H}}_{i,:}^{(l+1,a)} - \bar{\mathbf{H}}_{i,:}^{(l+1,b)}\|_2^2\right] \leq \mathbb{E}\left[\|\bar{\mathbf{H}}_{i,:}^{(l,a)} - \bar{\mathbf{H}}_{i,:}^{(l,b)}\|_2^2\right]. \tag{21}$$

We first relate distances between normalized representations to distances between the corresponding unnormalized ones. Since row-wise normalization is Lipschitz continuous on the set $\{\mathbf{z} : \|\mathbf{z}\|_2 \geq c\}$, we have

$$\left\|\frac{\mathbf{u}}{\|\mathbf{u}\|_2} - \frac{\mathbf{v}}{\|\mathbf{v}\|_2}\right\|_2 \leq \frac{1}{c}\|\mathbf{u} - \mathbf{v}\|_2.$$

Applying this inequality to $\mathbf{u} = \mathbf{H}_{i,:}^{(l,a)}$ and $\mathbf{v} = \mathbf{H}_{i,:}^{(l,b)}$ yields

$$\|\bar{\mathbf{H}}_{i,:}^{(l,a)} - \bar{\mathbf{H}}_{i,:}^{(l,b)}\|_2^2 \leq \frac{1}{c^2}\|\mathbf{H}_{i,:}^{(l,a)} - \mathbf{H}_{i,:}^{(l,b)}\|_2^2.$$

Thus, it suffices to show that the expected unnormalized difference between the two views contracts with message passing.

To this end, define the difference matrix at layer $l$ as $\boldsymbol{\Delta}^{(l)} \triangleq \mathbf{H}^{(l,a)} - \mathbf{H}^{(l,b)}$, we obtain

$$\boldsymbol{\Delta}^{(l+1)} = \mathbf{P}^{(a)}\mathbf{H}^{(l,a)} - \mathbf{P}^{(b)}\mathbf{H}^{(l,b)} = \mathbf{P}^{(a)}\boldsymbol{\Delta}^{(l)} + (\mathbf{P}^{(a)} - \mathbf{P}^{(b)})\mathbf{H}^{(l,b)}.$$

Taking expectation over the independent stochastic perturbations of the two views, the second term vanishes due to unbiasedness, i.e., $\mathbb{E}[\mathbf{P}^{(a)} - \mathbf{P}^{(b)}] = \mathbf{0}$. Moreover, since $\mathbf{P}^{(a)}$ is independent of $\boldsymbol{\Delta}^{(l)}$ in expectation and satisfies $\|\mathbf{P}^{(a)}\|_2 \leq 1$, we have

$$\mathbb{E}\|\boldsymbol{\Delta}^{(l+1)}\|_F^2 \leq \mathbb{E}\|\mathbf{P}^{(a)}\boldsymbol{\Delta}^{(l)}\|_F^2 \leq \mathbb{E}\|\boldsymbol{\Delta}^{(l)}\|_F^2.$$

Finally, when the anchor node $i$ is sampled uniformly,

$$\mathbb{E}_i\mathbb{E}\|\boldsymbol{\Delta}_{i,:}^{(l)}\|_2^2 = \frac{1}{N}\mathbb{E}\|\boldsymbol{\Delta}^{(l)}\|_F^2.$$

Combining the above inequalities shows that

$$\mathbb{E}\left[\|\bar{\mathbf{H}}_{i,:}^{(l+1,a)} - \bar{\mathbf{H}}_{i,:}^{(l+1,b)}\|_2^2\right] \leq \mathbb{E}\left[\|\bar{\mathbf{H}}_{i,:}^{(l,a)} - \bar{\mathbf{H}}_{i,:}^{(l,b)}\|_2^2\right],$$

which establishes (21) and thus $\mathbb{E}[R_{i,i}^{(l+1)}] \geq \mathbb{E}[R_{i,i}^{(l)}]$. $\qquad \square$

**Neighborhood-based positive samples.** In this case, we assume that positive samples are sampled uniformly from one-hop neighbors, i.e., $(v_i, v_j) \sim \text{Uniform}(\mathcal{T})$. We consider a single propagation operator $\mathbf{P}$ and $\mathbf{H}^{(l+1)} = \mathbf{P}\mathbf{H}^{(l)}$. Define $R_{ij}^{(l)} = \left\langle \bar{\mathbf{H}}_{i,:}^{(l)}, \bar{\mathbf{H}}_{j,:}^{(l)} \right\rangle$.

*Proof.* Similarly, we suffice to show that the expected normalized distance of a random edge decreases with $l$. For unit vectors, $R_{ij}^{(l)} = 1 - \frac{1}{2}\|\bar{\mathbf{H}}_{i,:}^{(l)} - \bar{\mathbf{H}}_{j,:}^{(l)}\|_2^2$. By the same normalization Lipschitz argument,

$$\|\bar{\mathbf{H}}_{i,:}^{(l)} - \bar{\mathbf{H}}_{j,:}^{(l)}\|_2^2 \leq \frac{1}{c^2}\|\mathbf{H}_{i,:}^{(l)} - \mathbf{H}_{j,:}^{(l)}\|_2^2.$$

Thus it is sufficient to prove the following edge-average contraction:

$$\mathbb{E}_{(v_i,v_j)\sim\mathcal{T}}\left[\|\mathbf{H}_{i,:}^{(l+1)} - \mathbf{H}_{j,:}^{(l+1)}\|_2^2\right] \leq \mathbb{E}_{(v_i,v_j)\sim\mathcal{T}}\left[\|\mathbf{H}_{i,:}^{(l)} - \mathbf{H}_{j,:}^{(l)}\|_2^2\right]. \tag{22}$$

Let $\mathbf{h}_{:,k}^{(l)}$ be the $k$-th feature dimension of $\mathbf{H}^{(l)}$. Using the definition of Dirichlet energy,

$$\sum_{(v_i,v_j)\in\mathcal{T}} \|\mathbf{H}_{i,:}^{(l)} - \mathbf{H}_{j,:}^{(l)}\|_2^2 = 2\sum_{k=1}^{D} \left(\mathbf{h}_{:,k}^{(l)}\right)^\top \mathbf{L}\,\mathbf{h}_{:,k}^{(l)}.$$

Because $\mathbf{H}^{(l+1)} = \mathbf{P}\mathbf{H}^{(l)}$ and $\|\mathbf{P}\|_2 \leq 1$, the Dirichlet energy is non-increasing under propagation, i.e.,

$$\left(\mathbf{P}\mathbf{h}_{:,k}^{(l)}\right)^\top \mathbf{L}\left(\mathbf{P}\mathbf{h}_{:,k}^{(l)}\right) \leq \left(\mathbf{h}_{:,k}^{(l)}\right)^\top \mathbf{L}\,\mathbf{h}_{:,k}^{(l)}, \quad \forall k.$$

Summing over $k$ yields

$$\sum_{(v_i,v_j)\in\mathcal{T}} \|\mathbf{H}_{i,:}^{(l+1)} - \mathbf{H}_{j,:}^{(l+1)}\|_2^2 \leq \sum_{(v_i,v_j)\in\mathcal{T}} \|\mathbf{H}_{i,:}^{(l)} - \mathbf{H}_{j,:}^{(l)}\|_2^2.$$

Dividing by $|\mathcal{T}|$ gives (22), and therefore $\mathbb{E}_{(v_i,v_j)\sim\mathcal{T}}[R_{ij}^{(l+1)}] \geq \mathbb{E}_{(v_i,v_j)\sim\mathcal{T}}[R_{ij}^{(l)}]$. $\qquad\square$

### E.2. Gradient view of the pre-alignment effect

We analyze how pre-alignment effect affects the learning signal of positive alignment under the InfoNCE objective in Eq. (4). For analytical clarity, we assume negative similarities unchanged, so as to isolate the effect of positive pre-alignment on the InfoNCE optimization.

**Theorem E.1.** *Consider the InfoNCE loss in Eq. (4), for any positive sample $p \in \mathcal{P}_i$, define $s_{ip} \triangleq \text{sim}(\mathbf{h}_i, \mathbf{h}_p)$, and $s_{in} \triangleq \text{sim}(\mathbf{h}_i, \mathbf{h}_n)$. Then the gradient magnitude w.r.t. the positive similarity satisfies:*

$$\left|\frac{\partial \mathcal{L}_{\text{InfoNCE}}(\mathbf{h}_i)}{\partial s_{ip}}\right| = \frac{1}{\tau} \cdot \underbrace{\frac{\exp(s_{ip}/\tau)}{\sum\limits_{p'\in\mathcal{P}_i} \exp(s_{ip'}/\tau)}}_{\text{positive weight}} \cdot \underbrace{\frac{\sum\limits_{n\in\mathcal{N}_i} \exp(s_{in}/\tau)}{\sum\limits_{p'\in\mathcal{P}_i} \exp(s_{ip'}/\tau) + \sum\limits_{n\in\mathcal{N}_i} \exp(s_{in}/\tau)}}_{\text{negative dominance factor}}. \tag{23}$$

*Assume a pre-alignment effect that increases all positive similarities by the same amount $\Delta \geq 0$, i.e., $s'_{ip} = s_{ip} + \Delta$ for all $p \in \mathcal{P}_i$, if $\mathcal{N}_i \neq \varnothing$, then for every $p \in \mathcal{P}_i$, we have:*

$$\left|\frac{\partial \mathcal{L}'_{\text{InfoNCE}}(\mathbf{h}_i)}{\partial s'_{ip}}\right| \leq e^{-\Delta/\tau} \left|\frac{\partial \mathcal{L}_{\text{InfoNCE}}(\mathbf{h}_i)}{\partial s_{ip}}\right|. \tag{24}$$

*Consequently, when message passing repeatedly increases positive similarity across layers (i.e., $\Delta = \Delta_t > 0$), the gradient signal provided by positive alignment diminishes rapidly as propagation depth grows.*

*Proof.* Define $A \triangleq \sum_{p\in\mathcal{P}_i} \exp(s_{ip}/\tau)$ and $B \triangleq \sum_{n\in\mathcal{N}_i} \exp(s_{in}/\tau)$. Then $\mathcal{L}_{\text{InfoNCE}}(\mathbf{h}_i) = -\log A + \log(A + B)$. For any $p \in \mathcal{P}_i$,

$$\frac{\partial \mathcal{L}_{\text{InfoNCE}}(\mathbf{h}_i)}{\partial s_{ip}} = -\frac{1}{A} \cdot \frac{1}{\tau} \exp(s_{ip}/\tau) + \frac{1}{A+B} \cdot \frac{1}{\tau} \exp(s_{ip}/\tau) = -\frac{1}{\tau}\frac{\exp(s_{ip}/\tau)\,B}{A(A+B)}.$$

Taking absolute values and rewriting gives Eq. (23).

For the pre-alignment effect $s'_{ip} = s_{ip} + \Delta$ for all positives, we have $A' = \sum_p \exp((s_{ip} + \Delta)/\tau) = e^{\Delta/\tau} A$ and $B' = B$. Also, the positive weight term in Eq. (23) is unchanged:

$$\frac{\exp((s_{ip} + \Delta)/\tau)}{\sum_{p'} \exp((s_{ip'} + \Delta)/\tau)} = \frac{e^{\Delta/\tau} \exp(s_{ip}/\tau)}{e^{\Delta/\tau} \sum_{p'} \exp(s_{ip'}/\tau)} = \frac{\exp(s_{ip}/\tau)}{\sum_{p'} \exp(s_{ip'}/\tau)}.$$

Therefore the ratio of gradient magnitudes reduces to the ratio of the "negative dominance factor":

$$\frac{\left|\partial \mathcal{L}'_{\text{InfoNCE}}/\partial s'_{ip}\right|}{\left|\partial \mathcal{L}_{\text{InfoNCE}}/\partial s_{ip}\right|} = \frac{B/(A' + B)}{B/(A + B)} = \frac{A + B}{e^{\Delta/\tau} A + B}.$$

Since $\Delta \geq 0$ implies $e^{\Delta/\tau} A + B \geq e^{\Delta/\tau}(A + B)$, we have

$$\frac{A + B}{e^{\Delta/\tau} A + B} \leq \frac{A + B}{e^{\Delta/\tau}(A + B)} = e^{-\Delta/\tau},$$

which proves Eq. (24). □

### E.3. Proof of Theorem 3.3

We first derive the bound under the commonly used local neighborhood-based positive sampling rules in graph contrastive learning, where positive samples correspond to neighboring nodes and negative samples are sampled independently from the marginal node distribution. We further discuss how the same proof strategy extends to the augmentation-based positive sampling at the end of the proof. For clarity, we introduce the setup used throughout the proof as follows:

Fix a propagation depth $l$ and a feature dimension $m$. Let $f \in \mathbb{R}^n$ denote the $m$-th feature across all nodes, where $f_i \triangleq x_{i,m}^{(l)}$. Let $\mathbf{T}$ be a lazy random-walk transition matrix, defined as $\mathbf{T} = \frac{1}{2}\mathbf{I} + \frac{1}{2}\mathbf{D}^{-1}\mathbf{A}$, where $\mathbf{A}$ is the adjacency matrix and $\mathbf{D}$ is the degree matrix. The laziness (i.e., staying at the current node with probability $1/2$) ensures that the random walk is aperiodic, which guarantees a strictly positive spectral gap. Let $\boldsymbol{\pi}$ denote the stationary distribution of $\mathbf{T}$, and let $\lambda_{\text{gap}} > 0$ be its spectral gap. We define the random-walk Laplacian as $\mathbf{L}_{\text{rw}} \triangleq \mathbf{I} - \mathbf{T}$, and define the Dirichlet energy of feature $f$ as

$$\mathcal{E}_m^{(l)} \triangleq f^\top \mathbf{L}_{\text{rw}} f = \frac{1}{2} \sum_{i,j} \boldsymbol{\pi}_i \, \mathbf{T}_{ij} \, (f_i - f_j)^2, \tag{25}$$

which measures the variation of the feature along edges of the graph under the random-walk distribution. We assume $f$ is centered with respect to $\boldsymbol{\pi}$, i.e., $\sum_i \boldsymbol{\pi}_i f_i = 0$.

We consider a binary random variable $S \in \{0,1\}$ indicating whether a node pair is a positive or a negative sample. Specifically, we sample $(i, j)$ as

$$(i, j) \sim \begin{cases} \mathbb{P}_+ : \ i \sim \boldsymbol{\pi}, \ j \sim \mathbf{T}(i, \cdot), & S = 1, \\ \mathbb{P}_- : \ i \sim \boldsymbol{\pi}, \ j \sim \boldsymbol{\pi} \text{ independently}, & S = 0, \end{cases} \tag{26}$$

which corresponds to local neighborhood–based positives and randomly sampled negatives. For the $m$-th feature dimension, we define the similarity score as $R_{ij,m}^{(l)} = \phi_m(f_i, f_j)$, where $\phi_m(a, b) = ab$ corresponds to the inner product, and the same form applies to cosine similarity after $\ell_2$ normalization. In both cases, we assume $|f_i| \leq B$ for all $i$, which implies the Lipschitz condition:

$$\left|\phi_m(f_i, f_j) - \phi_m(f_i, f_{j'})\right| \leq B \left|f_j - f_{j'}\right|. \tag{27}$$

Finally, to avoid imposing regularity assumptions on the score distribution, we apply a standard Gaussian smoothing. Define the observed score:

$$\widetilde{R}_{ij,m}^{(l)} \triangleq R_{ij,m}^{(l)} + \xi, \qquad \xi \sim \mathcal{N}(0, \sigma^2), \tag{28}$$

where $\xi$ is independent of $(S, i, j)$. Since $\widetilde{R}_{ij,m}^{(l)}$ is obtained from $R_{ij,m}^{(l)}$ by post-processing, we have:

$$I\left(S; R_{ij,m}^{(l)}\right) \leq I\left(S; \widetilde{R}_{ij,m}^{(l)}\right). \tag{29}$$

Therefore, it suffices to upper bound $I\left(S; \widetilde{R}_{ij,m}^{(l)}\right)$ in terms of $\mathcal{E}_m^{(l)}$.

*Proof.* Let $Q_1$ and $Q_0$ denote the conditional distributions of the similarity score $R_{ij,m}^{(l)}$ given $S = 1$ and $S = 0$, respectively. To compare these two distributions, we bound their Wasserstein-1 distance $W_1(Q_1, Q_0)$ by explicitly constructing a coupling. We first sample an anchor node $i \sim \pi$. Conditioned on $i$, we draw $j_+ \sim \mathbf{T}(i, \cdot)$ and $j_- \sim \pi$ independently, corresponding to a positive and a negative sample. We then define the coupled random variables

$$R_+ \triangleq \phi_m(f_i, f_{j_+}), \qquad R_- \triangleq \phi_m(f_i, f_{j_-}), \tag{30}$$

where $R_+$ and $R_-$ follow the distributions $Q_1$ and $Q_0$, respectively. We can get the following:

$$W_1(Q_1, Q_0) \leq \mathbb{E}[|R_+ - R_-|]. \tag{31}$$

We now bound the right side. Using the Lipschitz property in (27) and the Cauchy–Schwarz inequality, we obtain

$$\mathbb{E}[|R_+ - R_-|] \leq B \, \mathbb{E}[|f_{j_+} - f_{j_-}|] \leq B\sqrt{\mathbb{E}[(f_{j_+} - f_{j_-})^2]}. \tag{32}$$

Since $\mathbf{T}$ has stationary distribution $\pi$ and $i \sim \pi$, both $j_+$ and $j_-$ are marginally distributed according to $\pi$, and they are independent. Using the centering assumption $\sum_i \pi_i f_i = 0$, we have

$$\mathbb{E}[(f_{j_+} - f_{j_-})^2] = 2 \operatorname{Var}_\pi(f). \tag{33}$$

Combining the above inequalities yields

$$W_1(Q_1, Q_0) \leq B\sqrt{2 \operatorname{Var}_\pi(f)}. \tag{34}$$

Recall that $\mathbf{T}$ defines a lazy random-walk transition matrix. The laziness admits a strictly positive spectral gap $\lambda_{\text{gap}} > 0$. And we can use the Poincaré inequality to relate the variance of a function under the stationary distribution to its Dirichlet energy. Specifically, we have

$$\operatorname{Var}_\pi(f) \leq \frac{1}{\lambda_{\text{gap}}} f^\top L f = \frac{1}{\lambda_{\text{gap}}} \mathcal{E}_m^{(t)}. \tag{35}$$

Combining (34) and (35),

$$W_1(Q_1, Q_0) \leq B\sqrt{\frac{2}{\lambda_{\text{gap}}} \mathcal{E}_m^{(t)}}. \tag{36}$$

Since $\widetilde{R}_{ij,m}^{(l)} = R_{ij,m}^{(l)} + \xi$ is obtained by adding independent Gaussian noise $\xi \sim \mathcal{N}(0, \sigma^2)$. This smoothing ensures that the conditional distributions $\widetilde{Q}_1$ and $\widetilde{Q}_0$ of $\widetilde{R}_{ij,m}^{(l)}$ given $S = 1$ and $S = 0$ are absolutely continuous, allowing their KL divergence to be controlled.

$$D_{\text{KL}}(\widetilde{Q}_1 \| \widetilde{Q}_0) \leq \frac{1}{2\sigma^2} W_2^2(Q_1, Q_0) \leq \frac{1}{2\sigma^2} W_1^2(Q_1, Q_0), \tag{37}$$

where we used $W_2 \leq W_1$ for 1D distributions. Plugging (36) into (37) yields

$$D_{\text{KL}}(\widetilde{Q}_1 \| \widetilde{Q}_0) \leq \frac{1}{2\sigma^2} \cdot B^2 \cdot \frac{2}{\lambda_{\text{gap}}} \mathcal{E}_m^{(t)} = \frac{B^2}{\sigma^2 \lambda_{\text{gap}}} \mathcal{E}_m^{(t)}. \tag{38}$$

The same bound holds for $D_{\text{KL}}(\widetilde{Q}_0 \| \widetilde{Q}_1)$ by symmetry.

For binary $S$, we have:

$$I(S; \widetilde{R}) = \sum_{s \in \{0,1\}} \mathbb{P}(S = s) \, D_{\text{KL}}(\widetilde{Q}_s \| \widetilde{Q}), \qquad \widetilde{Q} = \sum_s \mathbb{P}(S = s) \widetilde{Q}_s. \tag{39}$$

$$I(S; \widetilde{R}) \leq \mathbb{P}(S = 1) D_{\text{KL}}(\widetilde{Q}_1 \| \widetilde{Q}_0) + \mathbb{P}(S = 0) D_{\text{KL}}(\widetilde{Q}_0 \| \widetilde{Q}_1) \leq \max\left\{ D_{\text{KL}}(\widetilde{Q}_1 \| \widetilde{Q}_0), D_{\text{KL}}(\widetilde{Q}_0 \| \widetilde{Q}_1) \right\}. \tag{40}$$

Combining with (38) gives

$$I(S; \widetilde{R}_{ij,m}^{(t)}) \;\leq\; \frac{B^2}{\sigma^2 \lambda_{\text{gap}}} \, \mathcal{E}_m^{(t)}.$$

Finally, by (29),

$$I(S; R_{ij,m}^{(t)}) \;\leq\; I(S; \widetilde{R}_{ij,m}^{(t)}) \;\leq\; \frac{B^2}{\sigma^2 \lambda_{\text{gap}}} \, \mathcal{E}_m^{(t)}.$$

Setting $C \triangleq \frac{B^2}{\sigma^2 \lambda_{\text{gap}}}$ proves (10).

**Remark (Augmentation-based positive sampling).** The above analysis also applies to the augmentation-based positive sampling rules. In this case, we define $S = 1$ to indicate that $(i, j)$ corresponds to the same node under two augmented views, and $S = 0$ to indicate two independent random nodes. The key observation is that the similarity score $R_{ij,m}^{(l)}$ remains a scalar function of the propagated features, and the randomness induced by view sampling only affects its conditional distribution. Consequently, the coupling construction in Step 1 and all subsequent bounds remain valid under the same boundedness and Lipschitz assumptions, yielding the same dependence on the Dirichlet energy $\mathcal{E}_m^{(l)}$. $\qquad\square$

### E.4. Proof of Theorem 3.4

*Proof.* Let $\mathbf{R}_{ij}^{(l)} = \big(R_{ij,1}^{(l)}, \dots, R_{ij,d}^{(l)}\big)$ and denote $\mathbf{R}_{\mathcal{H}} \triangleq \mathbf{R}_{ij,\mathcal{H}}^{(l)}$ and $\mathbf{R}_{\mathcal{L}} \triangleq \mathbf{R}_{ij,\mathcal{L}}^{(l)}$. We first prove (11). By the chain rule of mutual information,

$$I\big(S_{ij}; \mathbf{R}_{ij}^{(l)}\big) = I(S_{ij}; \mathbf{R}_{\mathcal{H}}) + I(S_{ij}; \mathbf{R}_{\mathcal{L}} \mid \mathbf{R}_{\mathcal{H}}). \tag{41}$$

Since conditional mutual information is nonnegative, we immediately have

$$I(S_{ij}; \mathbf{R}_{\mathcal{H}}) \leq I\big(S_{ij}; \mathbf{R}_{ij}^{(l)}\big), \tag{42}$$

which proves the left inequality in (11).

To prove the right inequality, it suffices to upper bound the increment $I(S_{ij}; \mathbf{R}_{\mathcal{L}} \mid \mathbf{R}_{\mathcal{H}})$. We adopt the following upper-bound condition: the incremental mutual information carried by the low-energy, conditioned on the high-energy, is bounded by the sum of the feature-wise mutual informations, i.e.,

$$I(S_{ij}; \mathbf{R}_{\mathcal{L}} \mid \mathbf{R}_{\mathcal{H}}) \leq \sum_{m \in \mathcal{L}} I\big(S_{ij}; R_{ij,m}^{(l)}\big). \tag{43}$$

Applying Theorem 3.3 to each $m \in \mathcal{L}$ gives

$$\sum_{m \in \mathcal{L}} I\big(S_{ij}; R_{ij,m}^{(l)}\big) \leq \sum_{m \in \mathcal{L}} C \, \mathcal{E}_m^{(l)} = C \sum_{m \in \mathcal{L}} \mathcal{E}_m^{(l)} \leq C \, \varepsilon, \tag{44}$$

where we used the assumption $\sum_{m \in \mathcal{L}} \mathcal{E}_m^{(l)} \leq \varepsilon$ in the last step. Combining (41), (43), and (44), we obtain

$$I\big(S_{ij}; \mathbf{R}_{ij}^{(l)}\big) \leq I(S_{ij}; \mathbf{R}_{\mathcal{H}}) + C \, \varepsilon, \tag{45}$$

which proves the right inequality in (11).

We now prove (12). Note that the scalar similarity $R_{ij}^{(l)} = \sum_{m=1}^{d} R_{ij,m}^{(l)}$ is a deterministic function of $\mathbf{R}_{ij}^{(l)}$. By the data processing inequality, applying any deterministic mapping cannot increase mutual information, hence

$$I\big(S_{ij}; R_{ij}^{(l)}\big) \leq I\big(S_{ij}; \mathbf{R}_{ij}^{(l)}\big). \tag{46}$$

This completes the proof. $\qquad\square$

*Table 9.* Node classification results on heterophilic graphs with MLP as base encoder (except SPGCL). Best results are in bold, and second-best are underlined. **X**, **A**, and **Y** denote node features, adjacency matrix, and node labels.

| Method | Training Data | Chameleon | Cornell | Texas | Wisconsin | Crocodile | Actor |
|---|---|---|---|---|---|---|---|
| MLP | **X, Y** | $49.06 \pm 1.86$ | $50.91 \pm 8.99$ | $58.92 \pm 4.56$ | $65.49 \pm 4.55$ | $64.41 \pm 0.53$ | $29.88 \pm 1.12$ |
| GRACE | **X, A** | $48.92 \pm 2.41$ | $64.59 \pm 6.30$ | $68.38 \pm 6.38$ | $\underline{80.00 \pm 2.41}$ | $65.03 \pm 1.18$ | $36.14 \pm 1.11$ |
| DGI | **X, A** | $42.74 \pm 1.80$ | $55.41 \pm 8.48$ | $60.27 \pm 5.70$ | $70.00 \pm 5.70$ | $63.62 \pm 0.96$ | $35.57 \pm 0.99$ |
| BGRL | **X, A** | $33.84 \pm 1.98$ | $45.95 \pm 4.77$ | $63.24 \pm 9.21$ | $49.80 \pm 8.02$ | $51.27 \pm 1.40$ | $28.47 \pm 1.40$ |
| GCA | **X, A** | $45.22 \pm 1.94$ | $50.00 \pm 7.00$ | $58.92 \pm 4.56$ | $52.16 \pm 5.16$ | $64.69 \pm 0.65$ | $35.03 \pm 0.91$ |
| LocalGCL | **X, A** | $\underline{59.39 \pm 1.56}$ | $30.81 \pm 8.66$ | $38.11 \pm 8.59$ | $41.18 \pm 7.34$ | $64.02 \pm 0.69$ | $29.47 \pm 0.78$ |
| GraphACL | **X, A** | $48.51 \pm 2.18$ | $\underline{69.19 \pm 4.63}$ | $\underline{72.70 \pm 5.32}$ | $77.65 \pm 2.95$ | $\underline{65.96 \pm 1.02}$ | $\underline{36.09 \pm 1.38}$ |
| **SPGCL(Ours)** | **X, A** | $\mathbf{72.26 \pm 1.66}$ | $\mathbf{75.41 \pm 6.43}$ | $\mathbf{80.81 \pm 6.30}$ | $\mathbf{83.53 \pm 4.26}$ | $\mathbf{77.43 \pm 0.67}$ | $\mathbf{37.23 \pm 1.19}$ |

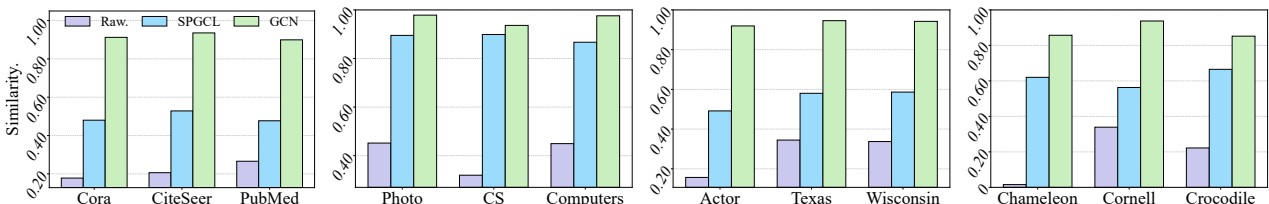

*Figure 4.* Average 1-hop neighbor similarity before contrastive optimization. Raw. means original node feature **X**.

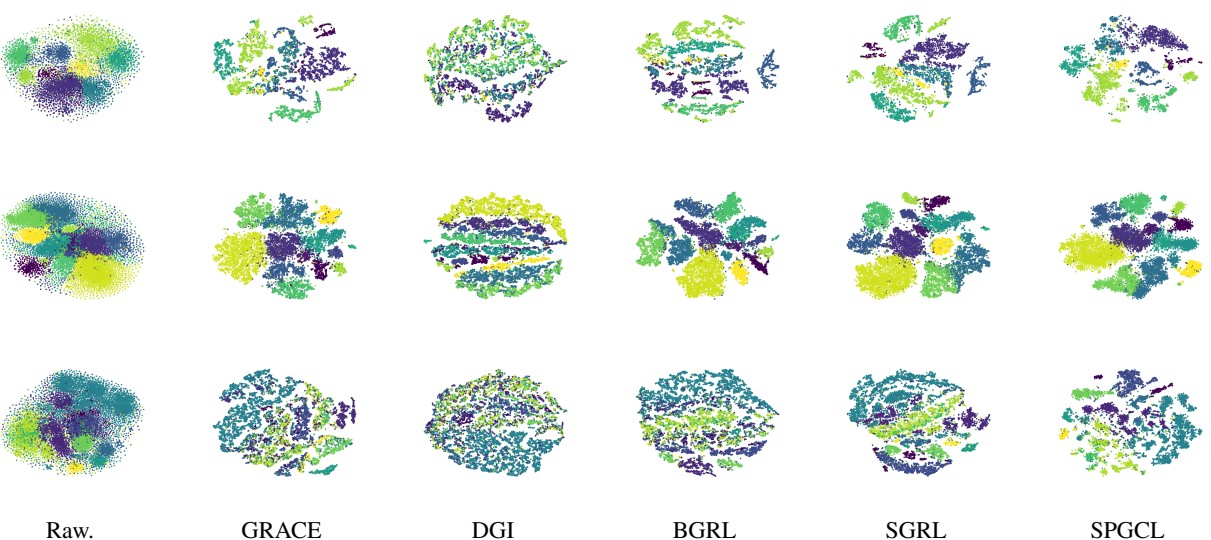

Raw.  GRACE  DGI  BGRL  SGRL  SPGCL

*Figure 5.* t-SNE embeddings of nodes on three datasets. From top to bottom: Photo, CS, and Computers.

# F. Additional Experiment Results.

### F.1. Visualization.

To better understand the quality of the learned representations, we visualize node embeddings with t-SNE (Van der Maaten & Hinton, 2008). Each point corresponds to a node, and various colors denote ground-truth classes. Figure 5 reports results on Photo, CS, and Computers. Compared with baselines, SPGCL yields more compact clusters within the same class. Class regions overlap less in many cases. The decision boundaries between classes also appear clearer. These patterns suggest that SPGCL learns embeddings with stronger class discrimination and better similar semantic grouping.

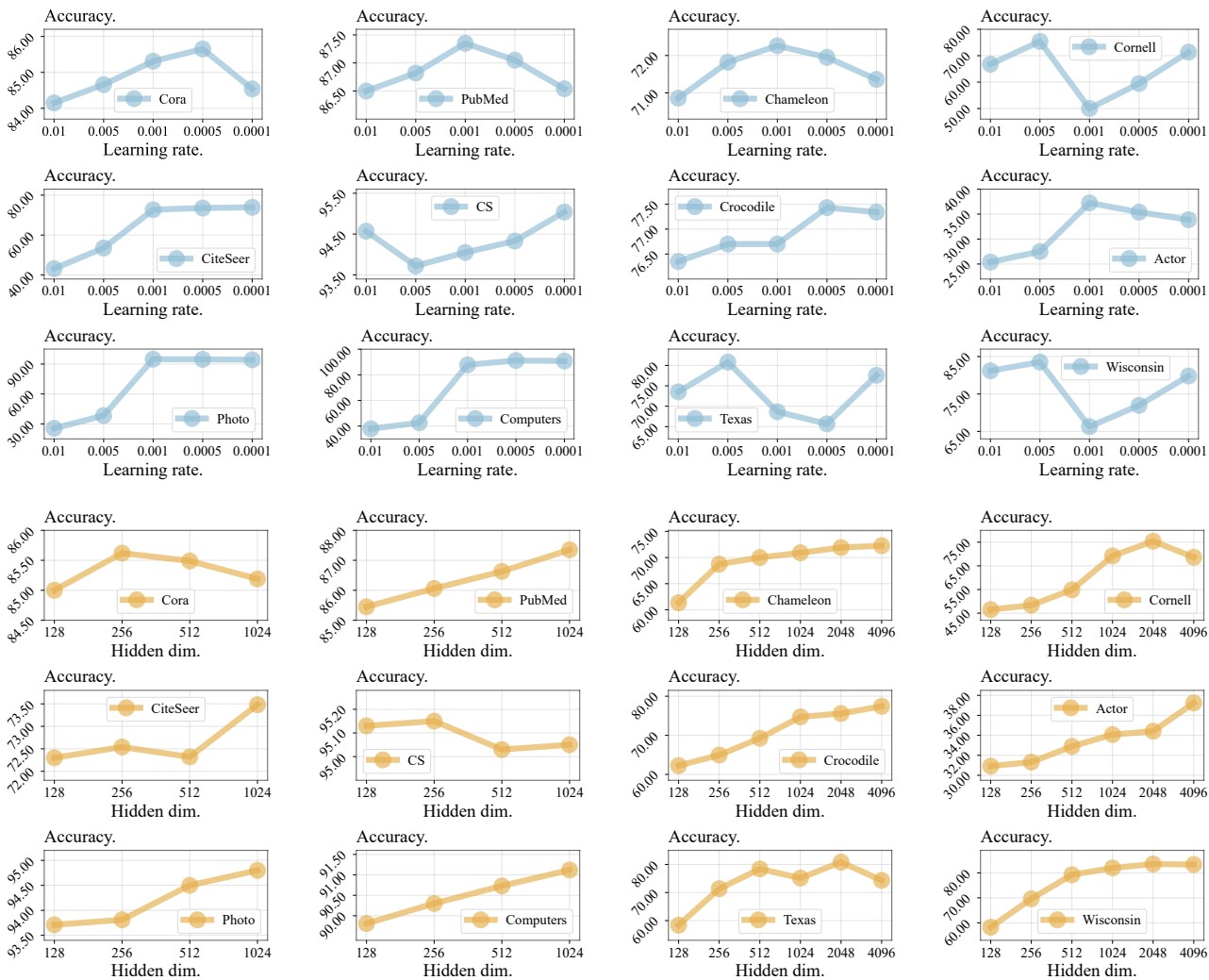

*Figure 6.* Hyper-parameter analysis of SPGCL with respect to the learning rate (blue) and the hidden dimension (orange). The left two columns correspond to homophilic graphs, while the right two columns correspond to heterophilic graphs.

## F.2. Empirical Analysis of the Pre-alignment Effect.

To empirically verify the pre-alignment effect induced by message passing, we further analyze the average similarity between each node and its one-hop neighbors. As shown in Figure 4, representations produced by a standard GCN exhibit a substantially higher neighbor similarity than the raw input features across all datasets, indicating that message passing alone can strongly increase the similarity of positive pairs even before contrastive optimization. In contrast, SPGCL consistently yields lower neighbor similarity than GCN. This confirms that SPGCL effectively mitigates the excessive pre-alignment caused by indiscriminate feature propagation, preserving sufficient diversity among positive pairs for contrastive learning.

## F.3. Impact of the base encoder choice.

To further validate the competitive performance of SPGCL, we replace the GNN encoder with MLP as the base encoder, given that nodes in heterophilic graphs are likely to belong to different classes despite being connected. As shown in Table 9, SPGCL still outperforms all baselines by a significant margin across all datasets. This highlights SPGCL's ability to effectively learn from positive samples.

*Table 10.* Performance comparison of dynamic and fixed partitions.

| Partition | Cora | CiteSeer | PubMed | Photo | CS | Computers |
|---|---|---|---|---|---|---|
| Dynamic | $85.97 \pm 0.82$ | $73.77 \pm 0.15$ | $87.36 \pm 0.34$ | $94.81 \pm 0.13$ | $94.78 \pm 0.05$ | $91.08 \pm 0.20$ |
| Fixed | $85.44 \pm 0.13$ | $73.49 \pm 0.26$ | $87.19 \pm 0.13$ | $94.83 \pm 0.07$ | $95.03 \pm 0.03$ | $91.04 \pm 0.09$ |

*Table 11.* Training time per epoch in seconds compared with InfoNCE-based methods.

| Method | Cora | CiteSeer | PubMed | Computers | Photo |
|---|---|---|---|---|---|
| SPGCL (Ours) | **0.0038** | **0.0043** | **0.0813** | **0.0774** | **0.0291** |
| GRACE | 0.0066 | 0.0071 | 0.1008 | 0.1068 | 0.0410 |
| GCA | 0.0062 | 0.0070 | 0.1311 | 0.1726 | 0.0638 |

## F.4. Learning rate and hidden dimension.

Figure 6 presents the node classification accuracy of SPGCL with different learning rate and hidden dimension on 12 datasets. For the learning rate, we observe a consistent unimodal pattern across most datasets. On homophilic graphs, performance improves as the learning rate increases and reaches its peak, after which it slightly degrades. Whereas on heterophilic graphs, performance is generally more sensitive to the learning rate, reflecting the noise induced by heterophilic neighbor mixing. For the hidden dimension, performance generally improves as the representation capacity increases. We notice that excessively large hidden dimensions may limit further gains, such as Texas, Cornell and Wisconsin.

## F.5. Effect of Dynamic Top-$K$ Partition.

In SPGCL, we adopt a fixed Top-$K$ feature partition before training, where feature dimensions are separated according to their Dirichlet energies computed from the input features. To examine whether dynamically updating the partition during training can further improve performance, we additionally compare the fixed partition with a dynamic Top-$K$ partition that recomputes the selected feature dimensions during training.

As shown in Table 10, the dynamic partition achieves comparable performance to the fixed one. It brings slight improvements on Cora, CiteSeer, PubMed, and Computers, while the fixed partition performs marginally better on CS and Photo. Overall, the performance difference between the two strategies is small. This suggests that the initial feature-energy-based partition already captures the main distinction between low-energy and high-energy feature dimensions. Since dynamic partitioning introduces extra computation at each training epoch, we use the fixed partition in SPGCL as it provides a better trade-off between effectiveness and efficiency.

## F.6. Efficiency Compared with InfoNCE-based Methods.

We compare the training time of SPGCL with representative InfoNCE-based methods, including GRACE and GCA. Unlike these methods that generate two augmented views and encode both views with GNNs, SPGCL follows a single-view contrastive design. The Dirichlet energy estimation is conducted only once before training, and the model uses one GCN branch with a lightweight MLP branch. As shown in Table 11, SPGCL consistently requires less training time per epoch than GRACE and GCA across all datasets. This demonstrates that SPGCL achieves a more efficient training process by avoiding the extra cost of dual-view augmentation and dual-view GNN encoding.

