# OpenReview forum: "Revisiting Positive Samples in Graph Contrastive Learning: From the Perspective of Message Passing"
_ICML.cc/2026/Conference — ICML 2026 regular_

### Official Review · Reviewer_umfa · 2026-02-25

**Soundness:** 3
**Presentation:** 3
**Significance:** 3
**Originality:** 3
**Overall Recommendation:** 4
**Confidence:** 4

**Summary:**

This paper re-examines the role of positive samples in Graph Contrastive Learning (GCL) through message passing. The authors observe that removing positive alignment from InfoNCE-based GCL often causes only minor performance drops, which they attribute to a *pre-alignment effect*: message passing increases similarity between positive pairs prior to contrastive optimization, weakening the effective learning signal.
To formalize this, the paper introduces **positive learning effectiveness**, defined as the mutual information between a binary positive indicator and embedding similarity. The authors prove that the feature-wise contribution to this mutual information is upper bounded by the feature’s Dirichlet energy. Since message passing reduces Dirichlet energy via smoothing, it limits the informative signal carried by positive pairs.
Motivated by this, the authors propose **SPGCL**, which separates features into high- and low-Dirichlet-energy subsets, propagating only high-energy features while using low-energy features for positive sampling. Experiments show consistent improvements over strong GCL baselines.

**Compliance With Llm Reviewing Policy:**

Affirmed.

**Final Justification:**

My concerns have been addressed. I maintain my positive score.

**Key Questions For Authors:**

### Questions

1. The analysis appears to overlook contrastive learning and instead studies message passing in isolation. Is this simplification practically justified? Could regularization on contrastive optimization itself counteract the pre-alignment effect?
2. How does SPGCL behave as GNN depth increases (e.g., in deeper or over-smoothing-prone settings)?

### Suggestions

1. Evaluate SPGCL with additional GNN backbones and varying depths.
2. Provide analysis or ablation on dynamic versus fixed TopK feature partitioning.
3. Include empirical studies correlating Dirichlet energy with measured positive learning effectiveness to validate Theorem 3.4 quantitatively.

**Limitations:**

yes

**Strengths And Weaknesses:**

### Strengths

1. The identification of the *pre-alignment effect* offers a clear and original explanation for the limited contribution of positive alignment in GCL.
2. The mutual information bound linking positive learning effectiveness to Dirichlet energy is insightful and well-motivated.
3. SPGCL directly follows from the theoretical findings.
4. Evaluation spans diverse datasets and tasks with consistent empirical gains.

### Weaknesses

1. It is unclear how tight or practically meaningful the bound is. The constant factors may be loose.
2. The theoretical analysis focuses on the encoder’s message passing dynamics while omitting the joint optimization of contrastive objectives.
3. Experiments primarily use GCN; generalization to other architectures (e.g., GAT, GraphSAGE) is not explored.
4. The TopK feature split is fixed before training; no analysis is provided on adaptive or dynamic alternatives.
5. The additional cost introduced by energy estimation and positive sampling is not thoroughly analyzed.

---

> ### Author Rebuttal · Authors · 2026-03-30
>
> We sincerely thank the reviewer for the time and effort.
>
> **R1 for W1**
>
> Thanks for the comment. We agree that the bound in the current paper is mainly intended to provide qualitative insight rather than a tight quantitative characterization. In particular, the constant factors may indeed be loose. Our goal is to show how message passing can weaken the effective learning signal from positive samples and to justify the motivation for separating feature components with different energies. We will clarify this point more explicitly in the revised version and avoid overstating the practical tightness of the bound.
>
> **R2 for W2 Q1**
>
> Thanks for the comment. We would like to clarify that we discuss the effect of the contrastive objective in Appendix E.2, Theorem E.1. There, under the InfoNCE loss, we analyze the gradient with respect to the positive similarity and show that when message passing increases the expected between positive pairs by a margin $\Delta \ge 0$, the corresponding gradient magnitude is upper-bounded by an exponentially decayed factor, i.e., it decreases by at least $e^{-\Delta/\tau}$. This result directly connects the pre-alignment effect to the optimization of the contrastive objective, by showing that stronger pre-alignment weakens the learning signal provided by positive alignment.
>
> **R3 for W3**
>
> Thanks for the comment. In the original submission, we mainly used GCN as the encoder because it is the standard backbone in graph contrastive learning and aligns with our analysis of message passing. To further address this concern, we additionally replaced the encoder with GAT and report the results below.
>
> | Encoder | Cora | Citeseer | PubMed | Computers | CS | Photo |
> |:-------:|:----:|:--------:|:------:|:---------:|:--:|:-----:|
> | GAT | 85.63±0.74 | 73.25±0.42 | 86.63±0.26 | 91.02±0.12 | 94.89±0.07 | 94.87±0.10 |
> | GCN | 84.44±0.13 | 73.49±0.26 | 87.19±0.13 | 91.04±0.09 | 95.03±0.03 | 94.83±0.07 |
>
> The results show that SPGCL remains competitive when using GAT, achieving performance comparable to GCN across the six datasets. These results suggest that the effectiveness of SPGCL is not restricted to GCN and can generalize to other encoder architectures. We will include this discussion more clearly in the revised version.
>
> **R4 for W4**
>
> Thanks for the comment. In the original submission, we used a fixed TopK split before training mainly for simplicity and efficiency. This design is also reasonable in our setting. In SPGCL, the propagated branch is $\mathbf{Z}_H=\sigma(\mathbf{P}\mathbf{X}_H\mathbf{W}_H)$, where $\mathbf{P}$ acts on the node dimension and is the source of the pre-alignment effect, while $\mathbf{W}$ acts on the feature dimension and mainly reweights or mixes the already selected channels. Therefore, the key decision is which input feature dimensions are allowed to enter message passing, and this is determined by their Dirichlet energies on the input features. In other words, the harmful effect comes from propagating overly smooth feature dimensions through $\mathbf{P}$ rather than from the later feature transformation itself. As a result, once the initial partition has separated obviously low-energy and high-energy dimensions, updating the partition dynamically during training brings only limited additional benefit in practice. To verify this, we further conducted experiments with dynamic repartition during training (**Please refer to Reviewer dhxx's R1 due to the limited rebuttal space**).
>
> **R5 for W5**
>
> Thanks for the comment. Please refer to **Reviewer 1LCb's R5** due to the limited rebuttal space.
>
> **R6 for Q2**
>
> Thanks for the comment. To examine the effect of GNN depth, we evaluate SPGCL with 2 to 10 GCN layers on three homophilic graphs (Cora, Citeseer, PubMed) and three heterophilic graphs (Wisconsin, Texas, Cornell).
>
> |Layers|Cora|Citeseer|PubMed|Wisconsin|Texas|Cornell|
> |:-------:|:--:|:------:|:----:|:-------:|:---:|:-----:|
> |2|85.43±0.15|73.45±0.27|86.48±0.10|83.73±4.03|80.81±7.58|74.05±5.72|
> |3|84.99±0.19|72.89±0.38|85.65±0.16|81.96±6.39|80.00±7.45|74.86±5.10|
> |4|84.67±0.30|72.34±0.24|85.00±0.24|82.16±4.57|77.83±4.18|73.24±5.16|
> |5|84.62±0.38|72.25±0.25|83.83±0.25|80.19±2.69|72.70±4.67|65.13±6.67|
> |6|84.54±0.32|72.17±0.41|83.13±0.37|60.30±6.19|70.00±4.84|48.10±4.55|
> |7|84.57±0.26|71.80±0.49|78.58±5.47|56.08±5.86|66.48±7.87|51.08±6.29|
> |8|84.17±0.33|71.52±0.20|59.18±18.06|64.31±4.51|64.86±10.50|56.75±6.62|
> |9|84.12±0.34|71.05±0.52|54.53±20.21|84.31±2.26|76.76±4.26|70.00±8.68|
> |10|83.84±0.29|70.91±0.22|53.93±19.22|81.57±5.33|80.27±2.86|71.35±7.34|
>
> On the homophilic graphs, the performance gradually decreases as the depth increases. This is consistent with the well-known over-smoothing effect. On the heterophilic graphs, the performance tends to decrease first and then increase as the depth grows. Overall,  SPGCL is still influenced by the depth of GCN.

---

> > ### Author Rebuttal · Reviewer_umfa · 2026-04-02
> >
> > Thank you for the detailed response and the additional experiments. Could you briefly explain why "On the heterophilic graphs, the performance tends to decrease first and then increase as the depth grows".

---

> > > ### Author Response · Authors · 2026-04-03
> > >
> > > Thanks for the valuable question. This phenomenon may be explained as follows.
> > >
> > > When the network is relatively shallow, SPGCL mainly benefits from selective propagation, where the propagated high-energy features preserve more discriminative information for positive learning. As the depth increases, the model aggregates more information from mismatched neighbors, and this can weaken the useful local signals captured in shallow layers, leading to a drop in performance. When the network becomes even deeper, the receptive field expands further, so node embeddings can access more global and long-range information. Since same-class nodes in heterophilic graphs are often connected through such longer-range patterns, this can become more helpful, which may explain the later recovery in performance.
> > >
> > > We will further investigate the behavior of SPGCL under over-smoothing settings in future work.

---

### Official Review · Reviewer_dhxx · 2026-03-10

**Soundness:** 3
**Presentation:** 3
**Significance:** 3
**Originality:** 3
**Overall Recommendation:** 5
**Confidence:** 4

**Summary:**

This paper investigates the role of positive samples in Graph Contrastive Learning (GCL). The authors motivate the paper from a counterintuitive phenomenon that GCLs can achieve competitive performance even without positive samples and attribute this phenomenon to the message passing mechanism, which inherently increases the similarity between positive samples, as theoretically proved using the Dirichlet energy. Based on this analysis, they propose Separate Propagation Graph Contrastive Learning (SPGCL), which decouples feature propagation into high-energy features, which are processed via GCN to preserve diversity, and low-energy features, which are transformed via MLP to guide probabilistic positive sampling. Extensive experiments demonstrate the effectiveness of SPGCL.

**Compliance With Llm Reviewing Policy:**

Affirmed.

**Final Justification:**

The authors‘ responses have addressed my comments. I maintain my positive score. I believe this paper makes a solid contribution towards graph self-supervised learning.

**Key Questions For Authors:**

See the aforementioned weaknesses. I have no further questions.

**Limitations:**

Yes

**Strengths And Weaknesses:**

Strengths:

S1. The motivation is interesting and insightful. The paper reveals a fundamental discrepancy between the widely assumed learning mechanism in GCL (using positive samples) and what empirically works during optimization (message passing itself), which brings new insights for positive samples utilization and GCL.

S2. The rigorous connection between Dirichlet energy and the mutual information of positive samples is technically sound. Most GCL papers focus on heuristic arguments, while the authors provide a principled bound showing that low-energy features upper-bound similarity but limit learning effectiveness.

S3. The Energy-Aware Propagation mechanism offers advantages over standard GCL designs. By separating features into different propagation paths, SPGCL addresses the observed problem and ensures that the contrastive objective can operate on actually effective feature components.

S4. The experimental results cover both homophilic and heterophilic graphs, providing convincing support for the proposed method.


Weaknesses:

W1. Regarding the proposed method, the Dirichlet energy is estimated once before training. However, a feature dimension that is high-energy at initialization might become smoother (low-energy) as the encoder learns, or vice versa. Fixing the partition based on input features might limit the model's adaptability. Can the feature energy partition be dynamically updated during training?

W2. The performance relies on selecting the Top-K high-energy dimensions. Figure 3 shows that performance is relatively stable on homophilic graphs but decreases as Top-K increases on heterophilic graphs. The optimal Top-K varies significantly across datasets. A more adaptive mechanism for determining the energy threshold could reduce the tuning burden.

W3. The final node representation is obtained by simple element-wise addition. It would be helpful if the authors could briefly explain whether alternative fusion mechanisms could be considered.

W4. Considering the technical depth of the paper, a notation table could be added to facilitate the understanding of the paper.

---

> ### Author Rebuttal · Authors · 2026-03-30
>
> We sincerely thank the reviewer for the time and effort.
>
> **R1 for W1**
>
> Thanks for the comment. We agree that the feature energy may change during training, and a fixed partition based on the input features may limit adaptability. In the original submission, we adopted a fixed partition for efficiency. To further examine this issue, we additionally conducted experiments with dynamic feature repartition during training below.
>
> |Partition | Cora | Citeseer | PubMed | Computers | CS | Photo |
> |:---:|:---:|:---:|:---:|:---:|:---:|:---:|
> | Dynamic | 85.97±0.82 | 73.77±0.15 | 87.36±0.34 | 91.08±0.20 | 94.78±0.05 | 94.81±0.13 |
> | Fixed | 85.44±0.13 | 73.49±0.26 | 87.19±0.13 | 91.04±0.09 | 95.03±0.03 | 94.83±0.07 |
>
> The results show that dynamic partitioning achieves performance comparable to the original static partitioning, but introduces extra computation at every epoch. These results suggest that the fixed partition used in SPGCL remains effective in practice, while offering a better trade-off between performance and efficiency.
>
> **R2 for W2**
>
> Thanks for the comment. We agree that the choice of Top-K affects performance, especially on heterophilic graphs. Our results suggest that heterophilic graphs are more sensitive because propagating too many feature dimensions may introduce more misleading signals. In the current version, Top-K is selected as a dataset-dependent hyperparameter. We agree that a more adaptive mechanism for determining the energy threshold would be valuable, as it could reduce tuning effort and improve robustness across datasets. We will highlight this limitation more clearly and explore adaptive thresholding in future work.
>
> **R3 for W3**
>
> Thanks for the comment. In the current version, we adopt element-wise addition as a simple and efficient fusion strategy to combine the complementary information from the two branches. This design is consistent with the goal of SPGCL, which is to leverage both propagated high-energy features and non-propagated low-energy features without introducing additional complexity. We agree that alternative fusion designs, such as learnable gating or attention-based fusion, are meaningful directions and may further improve the model. We will discuss this point in the revised version and leave a more advanced fusion mechanism for future work.
>
> **R4 for W4**
>
> Thanks for the comment. We agree that the notation in the current manuscript is relatively dense. Adding a notation table would improve readability and make the paper easier to follow. We will include a notation table in the revised version.

---

> > ### Author Rebuttal · Reviewer_dhxx · 2026-04-02
> >
> > The authors‘responses’ have addressed my comments. I maintain my positive score.

---

> > > ### Author Response · Authors · 2026-04-03
> > >
> > > Thanks very much for your time and effort!

---

### Official Review · Reviewer_1LCb · 2026-03-11

**Soundness:** 3
**Presentation:** 2
**Significance:** 3
**Originality:** 3
**Overall Recommendation:** 4
**Confidence:** 4

**Summary:**

This paper investigates why positive samples in Graph Contrastive Learning (GCL) provide surprisingly limited performance gains.
The authors empirically show that removing the positive alignment term from the InfoNCE loss barely degrades performance compared to standard GCL.
They attribute this to a pre-alignment effect: GNN message passing inherently increases cosine similarity between positive pairs before any contrastive optimization occurs, which trivializes the positive alignment objective.
Formalizing this through Dirichlet energy analysis, they prove (Theorem 3.3) that each feature dimension's contribution to positive learning effectiveness (defined as mutual information between a positive-indicator variable and the embedding similarity) is upper-bounded by its Dirichlet energy.
Low-energy features dominate similarity but carry negligible learning signal; high-energy features are where the informative signal resides.

Based on this analysis, the authors propose SPGCL (Separate Propagation Graph Contrastive Learning), which has two components:
* Energy-Aware Propagation (EAP): partition features by Dirichlet energy, propagate only high-energy features through a GCN, and process low-energy features via an MLP (no graph propagation);
* Energy-guided Positive Sampling (EPS): use low-energy features to compute a similarity matrix for Bernoulli-based positive pair selection, leveraging their stability to identify semantically reliable positives.

Experiments across 12 benchmarks (6 homophilic, 6 heterophilic) on node classification and clustering show consistent improvements over prior GCL methods.

**Compliance With Llm Reviewing Policy:**

Affirmed.

**Final Justification:**

The author addressed all my major concern and thus I maintain my positive score.

**Key Questions For Authors:**

* Can you provide the "w/o Pos." results on all 12 datasets? (Table 1, Table 3) The claim that positive samples are dispensable is central but currently supported by only three datasets (Figure 1).
* How does SPGCL perform when using decoupled GNN backbones (e.g., SGC, APPNP) instead of GCN? Since these architectures already separate propagation from transformation, they might naturally reduce the pre-alignment effect.
* What is the computational overhead of SPGCL relative to GRACE/GCA? The Dirichlet energy estimation is one-time, but the dual-path architecture (GCN + MLP) and the pairwise similarity matrix computation (Eq. 19, which is $O(N^2|L|)$) could be expensive on large graphs.
* Why is the EPS mechanism restricted to edges $(v_i, v_j) \in \mathcal{T}$? Have you experimented with sampling positives from k-hop neighborhoods or the full node set?

**Limitations:**

**Weakness**

**Suggestion for Clarity/Minor Errors**
* Line 026, Abstract: "theoretically finds" -> "theoretically find"
* Line 020: "intrinsic semantic and patterns" → "intrinsic semantics and patterns."
* Line 070, Section 1: "Why does positive samples fail" → "Why do positive samples fail."
* Line 113, Section 2: The notation $\mathcal{T}$ for edges is unconventional ($\mathcal{E}$ is standard);
* Line 271, Eq. 15: The symbol $\mathcal{H}$ is used for both the high-energy dimension set and the representation matrix $H^{(l)}$ earlier (Lemma 3).
This overloading is a bit confusing, especially in Section 3 where both appear.
* Figure 1: The figure should specify the evaluation metric (accuracy? F1?).
* Table 3: "heterphilic" → "heterophilic" (in the caption).
* Section 5.1: "Datsets" → "Datasets" (typo).
* The paper uses both "positive samples" and "positive pairs" interchangeably without clarifying the distinction. In GCL, a "positive sample" typically refers to the augmented view, while a "positive pair" is the (anchor, positive) tuple. Being precise would improve clarity.
* Eq. 20: The constraint $(v_i, v_j) \in \mathcal{T}$ means positives are restricted to edges. This is stated casually but is a significant design choice that limits the method to 1-hop neighbors. It should be discussed explicitly.
* The paper would benefit from a wall-clock time comparison table showing training time of SPGCL vs. baselines.
* Figure 3 (hyperparameter sensitivity): The x-axis uses a log scale for TopK but this is not explicitly stated.

**Strengths And Weaknesses:**

**Strength**
This paper is well-motivated and shares genuinely intriguing empirical findings. The observation that GCL without positive samples achieves competitive performance (Figure 1a) is striking and challenges a core assumption of the field.
This is a strong motivating experiment that naturally leads to the theoretical investigation.
The follow-up observation in Figure 1b that message passing itself inflates positive pair similarity, provides a clean empirical bridge to the theoretical story. This is the kind of finding that can redirect a research community's attention.

SPGCL's design is simple and principled as it follows directly from the theoretical analysis without introducing unnecessary complexity. The method does not require additional loss terms, learnable augmentation networks, or multi-stage training.
Additionally, the performance gains on heterophilic benchmarks are substantial and practically meaningful.

**Weakness**
* The "without positive samples" baseline (Figure 1a) is insufficiently characterized. Figure 1a is the paper's central motivating experiment, but it raises questions that are not adequately addressed:
  * What exactly is "w/o Pos."? The caption says "the positive alignment term is removed from the InfoNCE loss." This means the model is trained to push negatives apart without pulling positives together, which is essentially a uniformity-only objective. But uniformity alone is known to produce non-trivial representations (Wang & Isola, 2020, "Understanding Contrastive Representation Learning through Alignment and Uniformity"). The observation is therefore less surprising than the paper implies, it is partially explained by existing theory. This prior work should be discussed.
  * Only three datasets are shown. All three (Cora, CiteSeer, PubMed) are citation networks with high homophily. Does the same pattern hold on heterophilic graphs?

* The Dirichlet energy is computed on the raw input features X before training (Section 4.1: "This partition is computed once before training and remains fixed throughout optimization"). However, feature energies can change during training. The learned weight matrices $W_H$ and $W_L$ transform the features, potentially mixing high and low energy components in the hidden layers. A feature that is low-energy in $X$ might produce high-energy activations after $W_H$. The paper does not analyze whether the initial partition remains meaningful after weight updates.

* While EAP is directly motivated by the theory, EPS is motivated only by the heuristic argument that "low-energy features capture stable signals" (Section 4.2). Specifically, why is low-energy feature similarity a good proxy for semantic similarity? On heterophilic graphs, low-energy (smooth) features may reflect structural proximity, which the paper itself argues is unreliable for positive sampling in heterophilic settings. Also, the scaling hyperparameter $\alpha$ varies substantially across datasets (0.1 to 0.8 in Tables 7–8), suggesting the method is sensitive to this choice.

---

> ### Author Rebuttal · Authors · 2026-03-30
>
> We sincerely thank the reviewer for the time and effort.
>
> **R1 for W1 Q1**
>
> Thanks for the comment. The "w/o Pos." is a uniformity-only objective. Prior works have shown that such objective can still perform well in GCL, and a common explanation is that message passing implicitly induces positive alignment. We further revisits this and argue that message passing may hinder encoders from learning effectively from positive samples (Please refer to **Reviewer VZDR's R2** due to the limited space). We will clarify this more clearly in the revised version.
>
> We provide results on the remaining datasets below.
> |Method|Computers|CS|Photo|Chameleon|Cornell|Texas|Wisconsin|Crocodile| Actor|
> |:---:|:---:|:---:|:---:|:---:|:---:|:---:|:---:|:---:|:---:|
> |Random Init.|84.68±0.19|90.98±0.16|90.38±0.39|62.71±1.45|40.54±7.99|56.16±4.49|52.94±7.81|63.52±0.29|25.61±0.48|
> |InfoNCE|88.93±0.25|92.83±0.27|92.54±0.37 |64.66±1.20|42.84±7.73|60.57±5.29|58.98±6.03|71.25±0.44|29.86±0.76|
> |w/o Pos.|88.23±0.18|92.82±0.08|92.32±0.34 |64.36±1.34|42.24±7.59|60.27±4.78|57.82±5.32|70.58±0.66|29.37±0.62|
>
> The same phenomenon observed on homophilic graphs also holds on heterophilic graphs.
>
> **R2 for W2**
>
> Thanks for the comment. We conducted an experiment that dynamically repartitions features during training. The results are shown below.
> |Method|Cora|Citeseer|PubMed|Computers|CS|Photo|
> |:---:|:---:|:---:|:---:|:---:|:---:|:---:|
> |Dynamic|85.97±0.82|73.77±0.15|87.36±0.34|91.08±0.20|94.78±0.05|94.81±0.13|
>
> From the results, we observe that dynamic partitioning performs similarly to the original fixed partitioning. However, dynamic partitioning introduces additional computation. More detailed analysis please refer to **Reviewer umfa's R4**.
>
> **R3 for W3**
>
> Thanks for the comment. We agree that EPS is more heuristic compared with EAP. The reason is that, without labels, it is difficult to identify positive samples accurately. The design of EPS is inspired by SpCo [1], which suggests that low-frequency graph signals tend to be more stable. On heterophilic graphs, we adopt a smaller low-energy partition ratio to reduce the risk of incorrect positive samples. This reflects a limitation of SPGCL: it cannot adaptively determine the partition for different datasets. We will clarify this in the revised version. Regarding $\alpha$, it controls the retention probability of candidate positives, which helps improve training stability. The results when removing $\alpha$ are shown below.
> |Setting|Cora|Citeseer|PubMed|Computers|CS|Photo|Chameleon|Cornell|Texas|Wisconsin|Crocodile|Actor|
> |:----:|:----:|:--------:|:------:|:---------:|:--:|:-----:|:---------:|:-------:|:-----:|:---------:|:---------:|:-----:|
> |Removed|84.97±0.67|73.48±0.46|86.77±0.32|91.12±0.11|94.87±0.15|94.84±0.22|71.91±1.66|73.24±7.40|78.38±5.67|83.73±5.95| 77.19±0.61| 36.59±0.97|
> |SPGCL|85.44±0.13|73.49±0.26|87.19±0.13|91.03±0.09|95.03±0.03|94.83±0.07|72.26±1.66|75.41±6.43|80.81±6.30|83.53±4.26|77.43±0.67|37.23±1.19|
>
> Removing $\alpha$ leads to larger standard deviations, while the mean accuracy remains close to SPGCL.
>
>
> **R4 for Q2**
>
> Thanks for the comment. Although SGC and APPNP separate propagation from transformation, they still increase the similarity between positive pairs before optimizing, thus inducing the pre-alignment effect. Since SPGCL focuses on how to effectively learn from positive pairs during training, it remains effective when the encoder is replaced. The results with the encoder replaced by SGC are shown below.
> |Encoder|Cora|Citeseer|PubMed|Computers|CS|Photo|
> |:-----:|:--:|:------:|:----:|:-------:|:--:|:---:|
> |SGConv|85.76±0.78|73.88±0.75|86.94±0.31|91.29±0.17|94.94±0.06|94.16±0.11|
>
> The results of original GCN encoder are reported in the table above. Replacing it with SGConv still yields good performance.
>
> **R5 for Q3**
>
> Thanks for the comment. SPGCL has computational overhead comparable to, even lower than GRACE/GCA. SPGCL is a single-view contrastive approach. GRACE/GCA require generating two augmented views, encoding both views with GCN (SPGCL uses a GCN together with a lighter MLP). The Dirichlet energy estimation is performed only once before training. The results are shown below (training time per epoch in seconds).
>
> |Method|Cora|Citeseer|PubMed|Computers|Photo|
> |:----:|:--:|:------:|:----:|:-------:|:---:|
> |SPGCL|0.0038|0.0043|0.0813|0.0774|0.0291|
> |GRACE|0.0066|0.0071|0.1008|0.1068|0.0410|
> |GCA|0.0062|0.0070|0.1311|0.0726|0.0638|
>
> SPGCL requires less training time per epoch than GRACE/GCA on all datasets.
>
> **R6 for Q4**
>
> EPS is restricted to edges mainly for simplicity. We have not explored k-hop or full-set positive sampling. We will investigate this direction in future work.
>
> **R7**
>
> The evaluation metric of Fig.1 is Acc. Due to the limited rebuttal space, we are sorry that we can't respond to each point individually. We will carefully check the manuscript.
>
> [1] Revisiting graph contrastive learning from the perspective of graph spectrum. NeurIPS, 2022.

---

> > ### Author Rebuttal · Reviewer_1LCb · 2026-04-04
> >
> > Although some questions were deferred to future work, the authors' response addressed all of my major concerns.

---

> > > ### Author Response · Authors · 2026-04-04
> > >
> > > Thanks very much for your time and effort.

---

### Official Review · Reviewer_VZDR · 2026-03-16

**Soundness:** 3
**Presentation:** 3
**Significance:** 3
**Originality:** 3
**Overall Recommendation:** 4
**Confidence:** 3

**Summary:**

This paper revisits the role of positive samples in Graph Contrastive Learning. The authors uncover a counter-intuitive phenomenon: GCL models can still achieve competitive performance even when the positive sample alignment term is removed. Through theoretical analysis, the authors attribute this phenomenon to a "pre-alignment effect" induced by Message Passing. Based on this insight, they propose SPGCL , a method that separates feature propagation based on Dirichlet energy: high-energy features are propagated via GCN to preserve information diversity, while low-energy features are independently transformed via MLP and used to guide more reliable positive sample sampling. Experiments on 12 benchmark datasets  demonstrate that SPGCL outperforms existing state-of-the-art GCL methods on both node classification and clustering tasks.

**Compliance With Llm Reviewing Policy:**

Affirmed.

**Final Justification:**

The authors have addressed my concerns. Accordingly, I have raised my score.

**Key Questions For Authors:**

see weakness

**Strengths And Weaknesses:**

Strengths:

1. The paper provides a sound and rigorous theoretical analysis.
2. Experimental results demonstrate significant performance improvements of SPGCL on both homophilic and heterophilic graphs. This proves the method's broad applicability and practical value, holding promise for influencing future research directions regarding the mechanisms of graph self-supervised learning.

Weaknesses:

1. The authors fail to compare their method with state-of-the-art (SOTA) approaches from 2025.

2. To the best of my knowledge, prior works have already explored training without positive samples, such as SCE [1]. The authors should clearly articulate the differences between their work and this line of research.

[1] SCE: Scalable Network Embedding from Sparsest Cut. KDD, 2020.

3. The paper primarily focuses on node-level tasks (classification and clustering) and does not address link prediction or graph-level classification. Given that message passing and the definition of positive samples are equally critical in these tasks, the lack of evaluation on them limits our assessment of SPGCL's generalization capability.

---

> ### Author Rebuttal · Authors · 2026-03-30
>
> We sincerely thank the reviewer for the time and effort.
>
> **R1 for W1**
>
> Thanks for the comment. We add three recent SOTA GCL methods published in 2025 (EPAGCL [2], E2Neg [3], StrGCL [4]) and the representative GCL method SCE [1]. The results are shown below.
> |Method|Cora|Citeseer|PubMed|Computers|CS|Photo|Chameleon|Cornell|Texas|Wisconsin|Crocodile|Actor|
> |:----:|:--:|:------:|:----:|:-------:|:--:|:---:|:-------:|:-----:|:---:|:-------:|:-------:|:----:|
> |EPAGCL|84.40±0.58|71.00±0.77|86.60±0.13|89.91±0.29|93.07±0.07|92.93±0.25|64.08±2.21|41.62±3.86|56.22±2.48|54.71±4.75|69.50±0.89|29.32±1.01|
> |E2Neg|84.22±0.36|70.30±0.61|87.08±0.12|89.02±0.36|92.99±0.06|93.10±0.26|59.71±1.99|46.27±6.80|58.11±5.14|54.12±5.79|66.41±0.54|28.01±0.85|
> |StrGCL|84.86±0.54|72.49±0.11|86.52±0.19|90.32±0.05|94.06±0.03|93.94±0.12|62.21±1.52|48.11±5.38|64.32±7.31|60.59±5.22|63.06±1.16|24.98±0.84|
> |SCE|84.72±0.70|72.67±0.70|85.03±0.25|89.45±0.27|93.06±0.20|92.67±0.13|59.95±4.09|46.49±7.18|58.65±4.24|56.47±7.89|60.92±0.97|28.72±1.63|
> |SPGCL|85.44±0.13|73.49±0.26|87.19±0.13|91.03±0.09|95.03±0.03|94.83±0.07|72.26±1.66|75.41±6.43|80.81±6.30|83.53±4.26|77.43±0.67|37.23±1.19|
>
> On homophilic graphs, SPGCL performs better or on par with these baselines. On heterophilic graphs, SPGCL shows a larger advantage, outperforming baselines by an average margin of 18.51 percentage points across six datasets.
>
>
> **R2 for W2**
>
> Thanks for the comment. We agree that several prior studies [1, 8] have shown that GCL can remain effective without positive samples. A common view in these works is that message passing can partially take over the role of positive samples, which makes negative-only training feasible. We revisit this from a different perspective, that is, whether message passing may weaken the model’s ability to learn from positive samples. We agree that message passing can indeed drive neighboring nodes closer in the embedding space [1]. However, this effect occurs before the contrastive objective is optimized. As a result, it can weaken the model’s ability to learn effectively from positive samples. Based on this, we propose SPGCL to make fuller use of positive samples. In the revised version, we will clarify the difference between SPGCL and prior studies, and we will expand the related discussion and citations.
>
> **R3 for W3**
>
> Thanks for the comment. We add experiments on link prediction and graph classification. For link prediction, we follow the evaluation protocol in [5]. For graph classification, we use 10% labeled graphs to construct class prototypes, and then classify graphs based on similarity scores to these prototypes.
>
> |Link Prediction (Hits@100)|Cora|Citeseer|PubMed|Computers|Photo|
> |:---------------------:|:--:|:------:|:----:|:-------:|:---:|
> |GRACE|81.4±1.5|84.5±2.3|53.8±1.8|25.8±1.5|44.2±0.4|
> |DGI|73.7±2.0|78.7±2.5|53.0±2.6|23.2±2.6|44.1±3.4|
> |BGRL|71.1±2.5|74.7±1.1|57.5±1.9|24.2±2.5|38.0±1.2|
> |EPAGCL|72.7±3.2|81.6±4.0|53.9±2.1|25.1±0.6|49.2±1.3|
> |E2Neg|70.6±1.3|85.4±2.6|58.8±1.7|28.4±0.5|49.3±0.3|
> |SPGCL|85.0±2.0|92.1±1.2|72.7±1.1|32.1±1.8|52.5±3.4|
>
>
> |Graph Classification (Acc)|ENZYMES|COLLAB|IMDB-BINARY|PROTEINS|
> |:---------------------:|:-----:|:----:|:----------:|:------:|
> |GRACE|21.26±2.67|61.47±0.56|69.69±1.78|59.91±0.72|
> |DGI|21.10±2.79|63.47±0.90|68.74±1.55|55.66±0.85|
> |BGRL|20.31±3.01|63.79±0.61|63.43±2.08|60.81±1.37|
> |CaliGCL[6]|21.31±1.95|64.24±0.69|62.66±1.20|64.73±0.96|
> |Khan-GCL[7]|22.84±2.15|65.67±0.88|64.86±5.52|61.95±2.50|
> |SPGCL|31.73±1.84|75.24±0.86|72.19±1.24|72.62±0.82|
>
> |Graph Classification (F1)|ENZYMES|COLLAB|IMDB-BINARY|PROTEINS|
> |:--------------------:|:-----:|:----:|:----------:|:------:|
> |GRACE|16.68±3.10|61.52±0.54|69.59±1.82|55.71±0.80|
> |DGI|15.51±3.16|62.60±0.82|68.44±1.59|50.12±0.88|
> |BGRL|16.32±2.75|63.60±0.60|63.33±2.08|59.19±1.39|
> |CaliGCL[6]|17.38±2.53|60.19±0.86|62.94±1.53|60.67±1.02|
> |Khan-GCL[7]|20.03±2.46|64.51±0.98|62.07±8.00|60.24±2.39|
> |SPGCL|29.96±1.95|73.94±0.93|72.16±1.23|72.19±0.80|
>
> SPGCL performs well on both two tasks, indicating that the benefit of SPGCL is not limited to node-level tasks, but generalizes well to both edge-level and graph-level settings.
>
> [1] SCE: Scalable Network Embedding from Sparsest Cut. KDD, 2020.
>
> [2] Why does dropping edges usually outperform adding edges in graph contrastive learning? AAAI, 2025.
>
> [3] Does GCL need a large number of negative samples? Enhancing graph contrastive learning with effective and efficient negative sampling. AAAI, 2025.
>
> [4] Str-gcl: Structural commonsense driven graph contrastive learning. WWW, 2025.
>
> [5] Netinfof framework: Measuring and exploiting network usable information. Arxiv, 2024.
>
> [6] CaliGCL: Calibrated Graph Contrastive Learning via Partitioned Similarity and Consistency Discrimination. NeurIPS, 2025.
>
> [7] Khan-GCL: Kolmogorov–Arnold Network Based Graph Contrastive Learning with Hard Negatives. AAAI, 2026.
>
> [8] Architecture matters: Uncovering implicit mechanisms in graph contrastive learning. NeurIPS, 2023.

---

> > ### Author Rebuttal · Reviewer_VZDR · 2026-04-03
> >
> > Thank you for the detailed response. I have raised my score accordingly.

---

> > > ### Author Response · Authors · 2026-04-03
> > >
> > > Thank you for raising the score, and we sincerely appreciate the time and effort you devoted to the review process!

---

### Decision · Program_Chairs · 2026-04-30

**Decision:**

Accept (regular)

**Comment:**

After rebuttal and discussion steps, one reviewer raise the score and then all reviewers reach a consensus that this paper should be accepted.

This paper investigates the surprising limited contribution of positive samples in Graph Contrastive Learning and attributes it to a pre-alignment effect caused by message passing, formalized via Dirichlet energy analysis. Building on this insight, the authors propose SPGCL, which separates high- and low-energy feature propagation and uses low-energy features for energy-guided positive sampling. Anyway, the paper is well-motivated, technically sound, and validated across diverse benchmarks, providing new insights into positive sample utilization in GCL. Minor concerns include fixed feature partitioning, Top-K sensitivity, and potential computational overhead should be further refined and discussed in the final version.